# An ecological network approach for detecting and validating influential organisms for rice growth

**Masayuki Ushio[1,2,3]\*, Hiroki Saito[4], Motoaki Tojo[5], Atsushi J Nagano[6,7]**

[1]Hakubi Center, Kyoto University, Kyoto, Japan; [2]Center for Ecological Research, Kyoto University, Otsu, Japan; [3]Department of Ocean Science, The Hong Kong University of Science and Technology, Clear Water Bay, Kowloon, Hong Kong SAR, China; [4]Tropical Agriculture Research Front, Japan International Research Center for Agricultural Sciences, Okinawa, Japan; [5]Graduate School of Agriculture, Osaka Metropolitan University, Osaka, Japan; [6]Faculty of Agriculture, Ryukoku University, Otsu, Japan; [7]Institute for Advanced Biosciences, Keio University, Tsuruoka, Japan

## eLife assessment

There is a tremendous need to increase agricultural productivity with means that are both practical and efficient. Drawing on data from variable field environments, this **important** study provides a theoretical framework for the identification of new factors with presumed relevance for crop growth. This framework can be applied in the context of both agricultural and ecological studies. There is **solid** evidence for several of the authors' claims, but the impact of the study is limited due to missing functional validation of candidate species in the field. Plant biologists and ecologists working in agricultural and natural environments will find the work interesting.

\*For correspondence:
ong8181@gmail.com

**Abstract** How to achieve sustainable food production while reducing environmental impacts is a major concern in agricultural science, and advanced breeding techniques are promising for achieving such goals. However, rice is usually grown under field conditions and influenced by surrounding ecological community members. How ecological communities influence the rice performance in the field has been underexplored despite the potential of ecological communities to establish an environment-friendly agricultural system. In the present study, we demonstrate an ecological-network-based approach to detect potentially influential, previously overlooked organisms for rice (*Oryza sativa*). First, we established small experimental rice plots, and measured rice growth and monitored ecological community dynamics intensively and extensively using quantitative environmental DNA metabarcoding in 2017 in Japan. We detected more than 1000 species (including microbes and macrobes such as insects) in the rice plots, and nonlinear time series analysis detected 52 potentially influential organisms with lower-level taxonomic information. The results of the time series analysis were validated under field conditions in 2019 by field manipulation experiments. In 2019, we focused on two species, *Globisporangium nunn* and *Chironomus kiiensis*, whose abundance was manipulated in artificial rice plots. The responses of rice, namely, the growth rate and gene expression patterns, were measured before and after the manipulation. We confirmed that, especially in the *G. nunn*-added treatment, rice growth rate and gene expression pattern were changed. In the present study, we demonstrated that intensive monitoring of an agricultural system and the application of nonlinear time series analysis were helpful to identify influential organisms under field conditions. Although the effects of the manipulations were relatively small, the research framework presented here has future potential to harness the ecological complexity and utilize it in

agriculture. Our proof-of-concept study would be an important basis for the further development of field-basis system management.

## Introduction

Global food production supports energy requirements for human beings (*Godfray et al., 2010*), while the food production is a major driver of greenhouse gas emission and other environmental loads (*Aleksandrowicz et al., 2016*). How to achieve sustainable food production while reducing environmental impacts is thus a major concern in agricultural science (*Lenaerts et al., 2019*). Rice (*Oryza sativa*), one of the world's major staple crops, is an essential component of the diets and livelihood of over 3.5 billion people (*Wing et al., 2018*). A large number of studies have investigated how the performance of rice can be improved, and genomics-based breeding, combined with other advanced technologies such as high-throughput field-based phenotyping, is one of the most promising approaches to improve the performance (*Wing et al., 2018*).

While the advanced breeding techniques are promising, rice is usually grown under field conditions and inevitably influenced by the surrounding biotic and abiotic environment. Previous studies have investigated how meteorological and endogenous (e.g. plant age and genotype) variables influence the gene expression patterns (transcriptome dynamics) of rice under fluctuating field conditions (*Kashima et al., 2021*; *Nagano et al., 2012*), and have suggested that transcriptome dynamics can be predominantly explained by endogenous factors and abiotic variables such as ambient air temperature and solar radiation. Nonetheless, biotic variables such as insect herbivory and microbial mutualists/pathogens also play an important role in determining the transcriptome dynamics and productivity of crops (*Cohen and Leach, 2019*; *Savary et al., 2019*). However, the dynamics of biotic variables (i.e. ecological community members) under field conditions are difficult to predict because they often show more complex, nonlinear dynamics than abiotic variables (*Hsieh et al., 2005*). Thus, understanding whether and how biotic variables influence the rice performance (i.e. interspecific interactions) has been underexplored despite its importance for sustainable agriculture (*Toju et al., 2018*).

In ecological studies, biotic interactions have been studied theoretically and empirically for decades (*Allesina and Tang, 2012*; *May, 1972*; *Ushio et al., 2018a*). Ecologists have traditionally studied interspecific interactions based on observations and manipulations. For example, *Paine, 1966* measured the strength of interspecific interactions by removing predator in a rocky tidal system, and showed that the relatively less abundant predator species affected the community diversity disproportionally strongly through interspecific interactions (i.e. such a species is a so-called 'keystone' species). *Reynolds and Bruno, 2013* also showed that multiple predator species altered population growth of an estuarine food web through complex interactions among species. Thus far, interspecific interactions and their influences on the system dynamics have been studied as in *Wootton and Emmerson, 2005*. However, despite the substantial contributions to ecology, these observation- and manipulation-based approaches have critical limitations: the identification of multitaxa species and the quantification of their abundance under field conditions are challenging, and the quantification of their interactions is even more difficult (but see *Ushio, 2022*). Overcoming these difficulties and understanding how ecological community members influence the rice performance under field conditions will provide insights into how we can improve the rice performance and how rice responds to the ongoing and future anthropogenic impacts.

One of the promising approaches for overcoming the current limitations is to monitor the system frequently and detect interspecific interactions using time series data. Recent advances in empirical and statistical methods provide a practical way to achieve this goal. First, use of environmental DNA (eDNA) enables researchers to efficiently detect ecological community members under field conditions (*Taberlet et al., 2018*). Previous studies have shown that eDNA metabarcoding, an approach to comprehensively amplify and sequence DNAs belonging to a target taxa in an environmental sample, is a cost- and time-effective means to detect a large number of species (e.g. *Miya et al., 2015*), and the eDNA-based community data is especially informative when it is obtained quantitatively (e.g. sequencing with internal spike-in DNAs: *Ushio, 2022*; *Ushio et al., 2018b*). For example, quantitative eDNA metabarcoding enabled effective evaluation of intraspecific genetic diversity (*Tsuji et al., 2020*) and frequent and comprehensive monitoring of community dynamics (*Ushio, 2022*; *Ushio et al., 2023*). Second, nonlinear time series analytical tools enable researchers to reconstruct

complex interaction networks (*Chang et al., 2021*; *Deyle et al., 2016*; *Osada et al., 2023*; *Schreiber, 2000*; *Sugihara et al., 2012*), and previous studies adopting the approach have detected causality among many variables (e.g., reviewed in *Runge et al., 2019*). These methods detected and quantified biological interactions in complex systems such as microbiomes (*Chang et al., 2021*; *Fujita et al., 2023*), and contributed to understanding and forecasting complex dynamics driven by the interactions.

In the present study, we reconstructed the interaction network surrounding rice and detected potentially influential organisms for rice under field conditions. First, in 2017, we established small experimental rice plots, and monitored rice performance (i.e. growth rates) and ecological community dynamics intensively and extensively. Rice performance was quantified by measuring the growth rates (cm/day), and ecological community members were monitored by a quantitative eDNA metabarcoding approach (*Ushio, 2022*; *Ushio et al., 2018b*; *Figure 1*). We selected growth rates as a monitoring variable for rice because frequent and inexpensive monitoring is possible, and because it integrates various physiological states. The monitoring was performed daily from 23 May 2017 to 22 September 2017 (122 consecutive days). Second, we analyzed the generated extensive time series containing 1197 species and rice growth rates and produced a list of 52 potentially influential species using a time-series-based causality analysis (*Figure 2*). Third, in 2019, we empirically tested the effects of two species that had been identified as potentially influential in 2017 using manipulative experiments (*Figure 3*). During the growing season in 2019, an Oomycetes species, *Globisporangium nunn* (syn. *Pythium nunn*), was added, and a midge species, *Chironomus kiiensis*, was removed from small rice plots. The rice responses (the growth rate and gene expression patterns) were measured before and after the manipulation. We confirmed that the two species, especially *G. nunn*, indeed had statistically clear effects on the rice performance, which demonstrated that integration of the eDNA-based monitoring and time series analysis would be effective for detecting previously overlooked influential organisms in agricultural systems.

## Results

### Field monitoring of rice growth and ecological communities in 2017

In five rice plots established in 2017 in an experimental field at Kyoto University, Japan, daily rice growth rate (cm/day in height) was monitored during the growing season by measuring rice leaf height of target individuals (red points in the right panel of *Figure 1a*) every day using a ruler, which showed consistent patterns among the plots (*Figure 1b*). Daily mean air temperature also showed consistent patterns among the five plots (*Figure 1c*). The daily growth rate reached the maximum during late June to early July, and the height of rice individuals did not increase after the middle of August (first headings appeared on 12 or 13 August in the five plots). During the monitoring period, we occasionally observed decreases in the rice heights due to mechanical damage or insect herbivores, but it is unlikely that they affected our causal inferences because the changes in the rice heights due to the damages and herbivories were smaller and less frequent than those due to growth. See *Figure 1—figure supplement 1* and a video (https://doi.org/10.6084/m9.figshare.19029650.v1) for additional information regarding the monitoring.

Daily ecological community dynamics was monitored using quantitative eDNA metabarcoding by analyzing eDNA in water samples taken from the five plots with four universal primer sets (16 S rRNA, 18 S rRNA, ITS, and COI regions targeting prokaryotes, eukaryotes, fungi, and animals, respectively; see Materials and methods), and detailed patterns of the ecological community dynamics and their implications were reported in *Figure 1—figure supplement 2*, a video (https://doi.org/10.6084/m9.figshare.23514150.v1), and *Ushio, 2022*. Briefly, the total eDNA copy number increased late in the sampling period (*Figure 1d*). In contrast, ASV diversity (a surrogate of species diversity) was highest in August and then decreased in September (*Figure 1e*). Prokaryotes largely accounted for this pattern (*Figure 1d*). In the previous study, a large ecological interaction network was reconstructed (*Figure 1—figure supplement 1c*), and possible mechanisms of the ecological dynamics were discussed in detail (*Ushio, 2022*). Importantly, the time series-based causality analysis used in the next section requires quantitative time series, and our quantitative eDNA time series are suitable for this purpose.

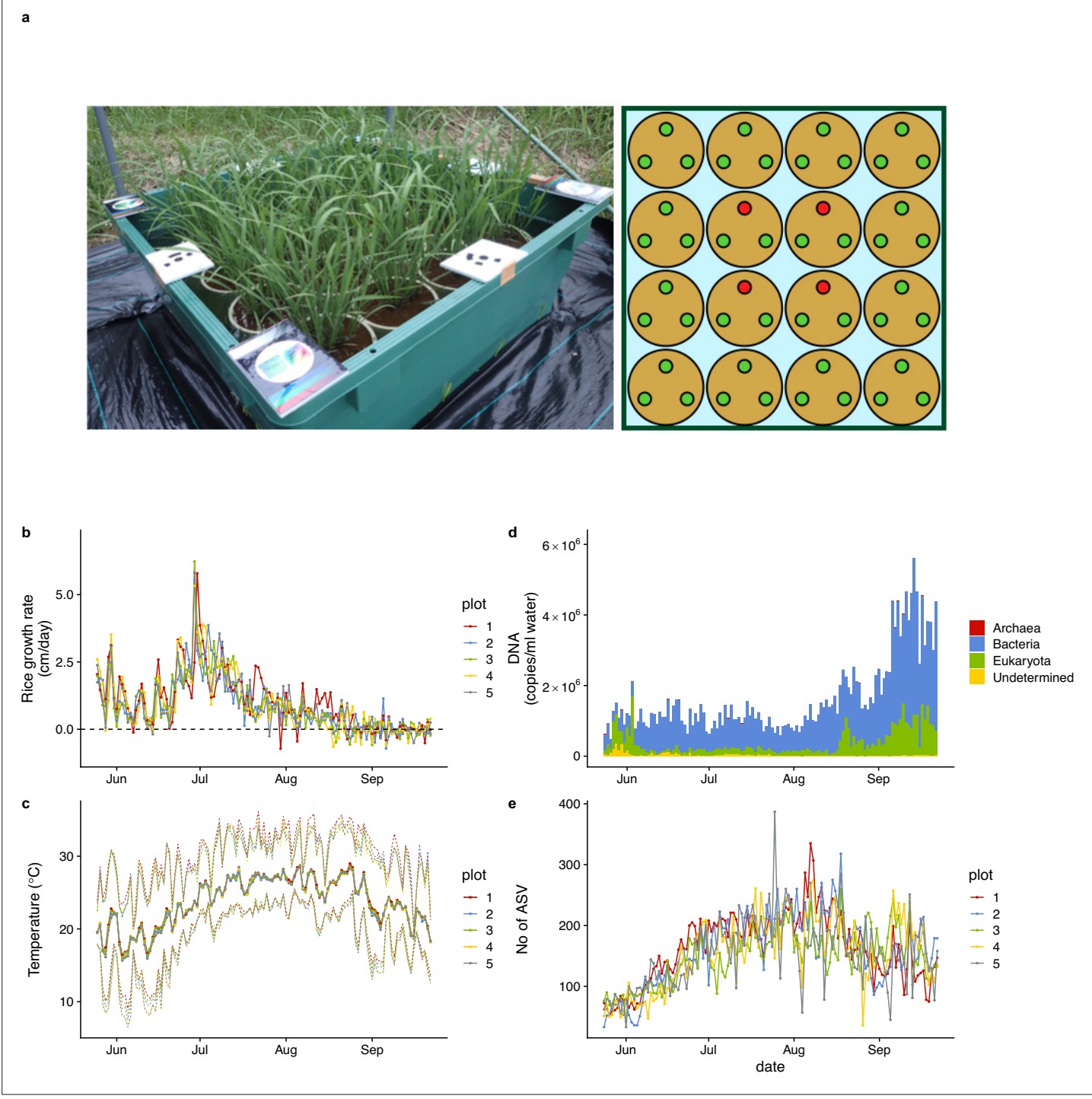

**Figure 1.** Rice plot, growth rate, air temperature, and ecological community dynamics. (**a**) 90 cm × 90 cm rice plot and Wagner pot alignments. Three rice individuals were grown in each pot. Heights and SAPD of the four red individuals in each plot were measured every day during the monitoring period, and the average values of the four individuals were regarded as representative values for each plot. (**b**) Rice growth rate (cm/day). (**c**) Daily mean air temperature measured at each rice plot. Upper and lower dotted lines indicate daily maximum and minimum air temperature. (**d**) Ecological community compositions and average DNA copy numbers per ml water (reported in *Ushio, 2022*). (**e**) The number of amplicon sequence variants (ASVs) from each water sample (reported in *Ushio, 2022*). For (**b**), (**c**), and (**e**), different colors indicate data from different rice plots.

The online version of this article includes the following figure supplement(s) for figure 1:

**Figure supplement 1.** Monitoring framework of ecological community (as in *Ushio, 2022*).

**Figure supplement 2.** Environmental DNA (eDNA)-based monitoring of ecological communities in 2017.

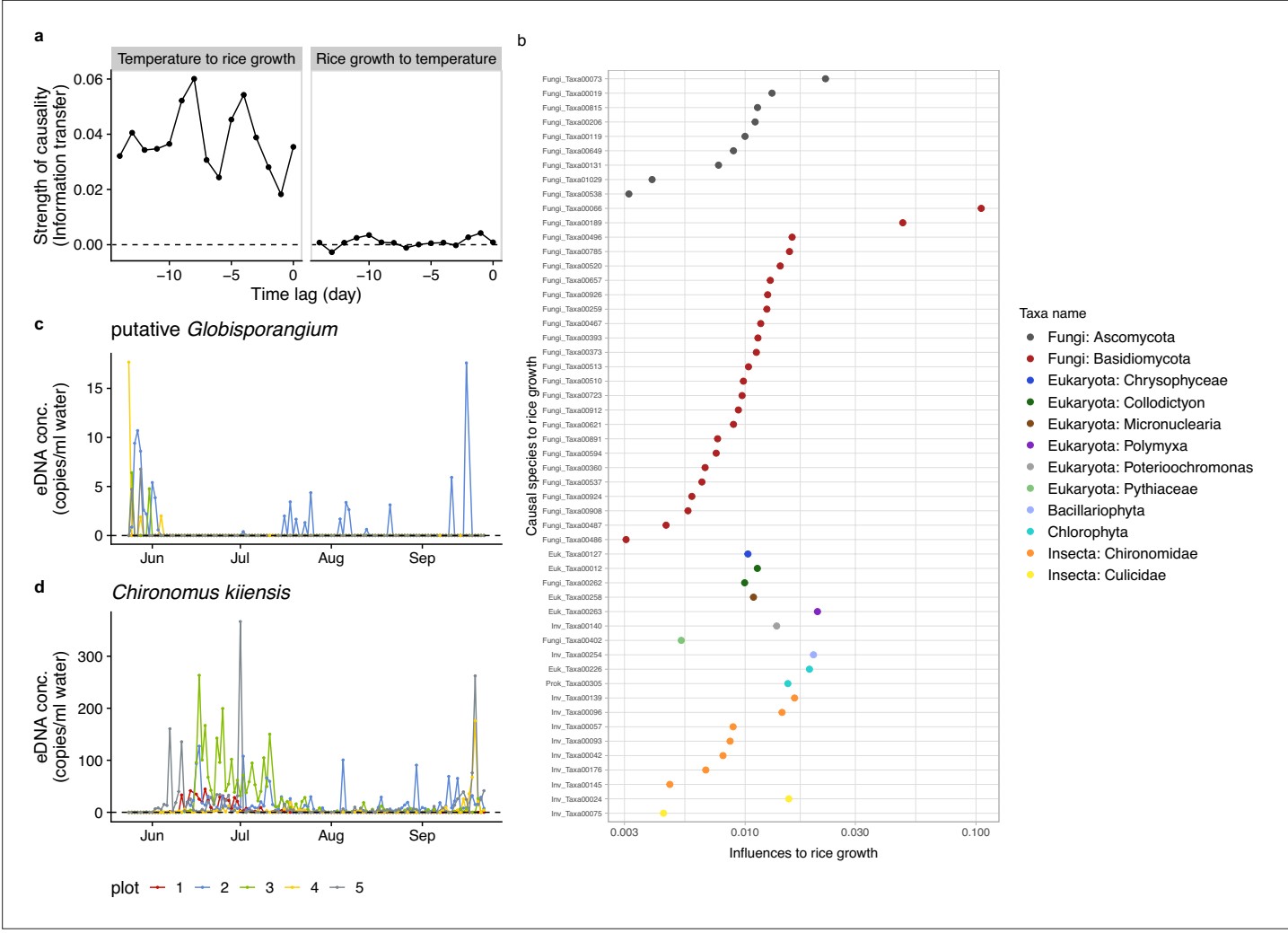

**Figure 2.** Information transfer between rice growth and ecological community members. (**a**) An example of the results of the unified information-theoretic causality (UIC) analysis. Information transfer between air temperature and rice growth rates was quantified. Much higher information transfer was detected from air temperature to rice growth (left panel) compared with the opposite direction (right panel). (**b**) Strength of causal influence from ecological community members to rice growth. Transfer entropy (TE) quantified by the UIC method was used as an index of causal influence. Colors indicate taxa assigned to ASVs. *y*-axis indicates ASV ID. Note that the prefix (e.g. 'Fungi_') of the IDs corresponds a major target group of the primer and does not necessarily indicate a taxonomic group assigned to the ASV (see *Supplementary file 1*). (**c**) eDNA dynamics of putative *Globisporangium nunn* (Fungi_Taxa00402 in *Supplementary file 1*). (**d**) eDNA dynamics of *Chironomus kiiensis* (total DNA copy numbers of five midge ASVs). For (**c**) and (**d**), different colors indicate data from different rice plots.

## Detection of potential causal relationships between rice growth and ecological community dynamics

Potential causal relationships among the daily rice growth rates, eDNA time series, and climate data were detected using unified information-theoretic causality (UIC) analysis, which quantifies information flow in terms of transfer entropy (TE; *Figure 2*; for UIC, see '*Detection of potential causal relationships between rice growth and ecological community members*' in Materials and methods and *Osada et al., 2023*). The information flow between eDNA time series and rice growth rates was regarded as a sign of interactions between rice and the ecological community member. *Figure 2a* is a demonstration of the performance of the UIC analysis, showing the influence of air temperature on the rice growth. The left panel in *Figure 2a* indicates that there were strong effects of air temperature on the rice growth (*x*-axis indicates the time-lag of the effects), while the right panel indicates that there were virtually no effects of the rice growth on air temperature, which is biologically convincing. In total, 718 ASVs were identified as potentially causal (i.e. statistically clear information flow; $p<0.05$; we

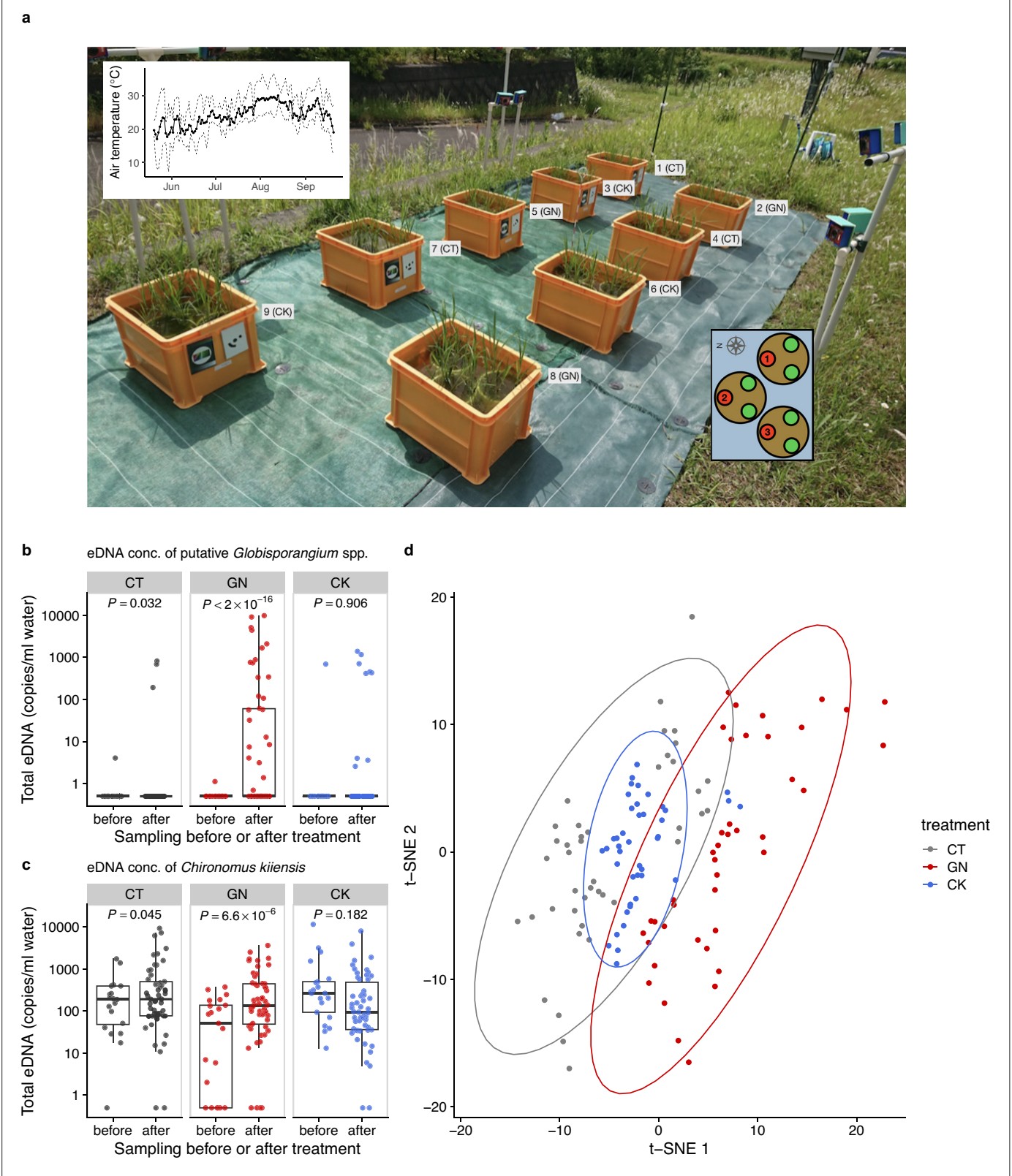

**Figure 3.** The manipulation experiment performed in 2019 and ecological community compositions before and after the manipulation. (**a**) Setting of the manipulation experiment in 2019. The number and characters next to each plot indicate the plot number and treatment. The left-top inset shows daily mean air temperature (thick line) and daily maximum and minimum air temperature (dashed lines). The right-bottom inset shows three individuals (red and green points) in each Wagner pot, and the number in each red individual indicates the pot location number. Heights and SPAD

*Figure 3 continued on next page*

*Figure 3 continued*

of the red individuals were measured. Total eDNA copy numbers of (**b**) putative *Globisporangium* spp. and (**c**) midge (*Chironomus kiiensis*) in the rice plots. 'before' and 'after' indicate 'from 18 June to 24 June' and 'from 25 June to 12 July', respectively. (**d**) Overall community compositions after the manipulation. Gray, red, and blue indicate CT (control), GN (*Globisporangium nunn* added), and CK (*Chironomus kiiensis* removed) treatments, respectively.

The online version of this article includes the following figure supplement(s) for figure 3:

**Figure supplement 1.** *Globisporangium nunn* and Midge (*Chironomus kiiensis*) used in the manipulation experiment in 2019.

**Figure supplement 2.** Environmental DNA (eDNA)-based monitoring of ecological communities in 2019.

use the term 'statistical clarity' instead of 'statistical significance', according to the recommendation by *Dushoff et al., 2019*), among which 52 ASVs were identified with lower-level taxonomic information (*Supplementary file 1*). The potentially causal ASVs belong to various taxa, including fungi and insects, with varying degree of the effects, measured as the information flow (*Figure 2b*). Except for one species (*Penicillium citrinum*), we could not find information regarding the effects of the detected ASVs on rice performance, at least at the time of this analysis.

Among the 52 ASVs, we particularly focused on two ASVs, *Globisporangium nunn* (Oomycete; previously known as *Pythium nunn*) (*Uzuhashi et al., 2010*) and *Chironomus kiiensis* (midge species; see the eDNA dynamics of putative *G. nunn* and *C. kiiensis* in *Figure 2c and d*), in the field manipulation experiments in 2019. The reasons why we focused on these two species in 2019 are multifold. First, they can be relatively easily manipulated; *G. nunn* had already been isolated and cultivated (*Kobayashi et al., 2010*; *Tojo et al., 1993*) and *C. kiiensis* is an insect species and relatively large (several mm to cm). Second, the two species had not been reported to be pathogens; using pathogen species under field conditions is not recommended in the experimental field because they might have adverse effects on nearby agricultural systems. *G. nunn* is a potential biocontrol agent and is reported to be beneficial for vegetables (*Kobayashi et al., 2010*). *C. kiiensis* is a common midge species around the study region (including in rice paddy fields). The larvae are usually about 10 mm and adults are about 5 mm. Additional information for the two species is described in Materials and methods and *Supplementary file 2* (see also 'Notes on *Globisporangium nunn*').

## The ecological community composition after the manipulation experiments in 2019

In 2019, we established nine experimental plots to monitor rice growth and to perform field manipulation experiments in the same experimental field. The plots were similar to, but smaller than, those used in 2017 so that the number of replicates could be increased (*Figure 3a*). We prepared three treatments, namely, the control treatment (CT), *G. nunn*-added treatment (GN), and *C. kiiensis*-removed treatment (CK), and there were three replicates for each treatment (3 treatments ×3 replicates = 9 plots). We manipulated the abundance of *G. nunn* and *C. kiiensis* by adding *G. nunn* and removing *C. kiiensis*, respectively (*Figure 3—figure supplement 1*), and the manipulation experiments were performed three times (on 24, 26, and 28 June 2019). We quantified the eDNA dynamics of the ecological communities weekly or biweekly except during the intensive, daily monitoring period before and after the field manipulation experiments (from 18 June to 12 July; see the sampling design in *Supplementary file 3* and overall eDNA dynamics in *Figure 3—figure supplement 2*). As for the GN treatment, we detected 19 putative *Globisporangium* sequences belonging to the family Pythiaceae from the four marker genes (i.e. 16 S, 18 S, ITS and COI), most of which we could not identify with lower-level taxonomic information partially because of the conservative assignment algorithm of the sequence analysis pipeline, Claident (*Tanabe and Toju, 2013*). We described them as 'putative *Globisporangium* spp.' and regarded them as *G. nunn* that we added because the eDNA concentrations of these Pythiaceae sequences were increased after we added *G. nunn* isolates.

The increase in the eDNA concentrations of putative *Globisporangium* spp. in the GN treatment after the manipulation was statistically clear ($p < 2.0 \times 10^{-16}$; *Figure 3b*). In addition, there was a slight, but statistically clear increase in the eDNA concentrations of putative *Globisporangium* spp. in the CT treatment after the manipulation ($p = 0.032$; *Figure 3b*). On the other hand, the eDNA concentrations of putative *Globisporangium* spp. in the CK treatment was not changed after the manipulation ($p > 0.05$). The eDNA concentrations of *C. kiiensis* were statistically clearly changed after the

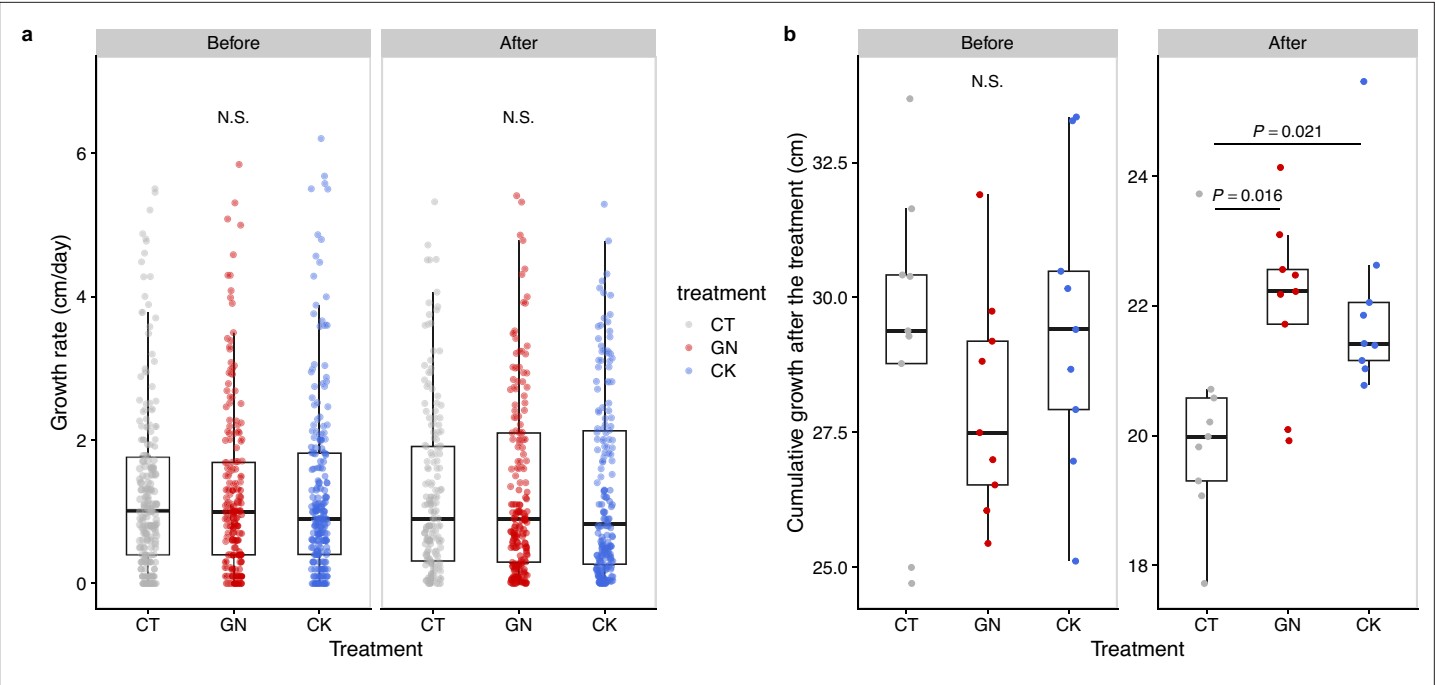

**Figure 4.** Rice growth rate and cumulative growth before and after the manipulation experiment in 2019. (**a**) Growth rates and (**b**) cumulative growth of the rice individuals in the three treatments (CT = control; GN = *Globisporangium* nunn added; CK = *Chironomus kiiensis* removed) before and after the manipulation (cumulative growth was calculated by summing up height growth before the third manipulation or during 10 days after the third manipulation).

The online version of this article includes the following figure supplement(s) for figure 4:

**Figure supplement 1.** Rice growth trajectory in 2019.

manipulation in the CT (slightly decreased; p=0.045; *Figure 3c*) and GN (increased; p=6.6 × 10⁻⁶; *Figure 3c*) treatments. In the CK treatment, the eDNA concentration of *C. kiiensis* was not changed (p>0.05; *Figure 3c*). These results were further visualized and confirmed by the *t*-distributed stochastic neighbor embedding (t-SNE) (*Van der Maaten and Hinton, 2008*) analysis of the overall community composition. Ecological communities of the GN treatment were generally separated from those of the CT treatment, while those of the CK treatment almost overlapped with those of the CT treatment (*Figure 3d*). Alternative statistical modeling that included the treatments (the control versus GN or CK treatments) and manipulation timing (i.e. before or after the manipulation), which simultaneously took the temporal changes of all the treatments into account, also showed qualitatively similar results (*Supplementary file 4*), further supporting the results.

## Rice performance after the field manipulation experiments in 2019

We measured the rice growth as in 2017 and compared those of the GN and CK treatments with that of the CT treatment before and after the manipulation experiments. In general, we did not find statistically clear differences among the treatment in the daily rice growth rate before and after manipulation (p>0.05; *Figure 4a*; see *Figure 4—figure supplement 1* for the rice growth trajectory during the growing season). However, we found that the cumulative rice growth (during 10 days after the manipulation) increased in the GN and CK treatments after the manipulation, and the effect was statistically clear (p=0.016 and p=0.021 for the GN and CK treatments, respectively; *Figure 4b*). Alternative statistical modeling that included the treatments (the control versus GN or CK treatments) and manipulation timing (i.e. before or after the manipulation), which simultaneously took the temporal changes of all the treatments into account, also showed qualitatively similar results (Materials and methods and *Supplementary file 5*). We did not find any statistically clear difference in the rice yields (e.g. the number of rice grains per head, the grain weights, or the proportion of fertile grains) among the treatments (see *Supplementary file 6* for the raw rice yield data).

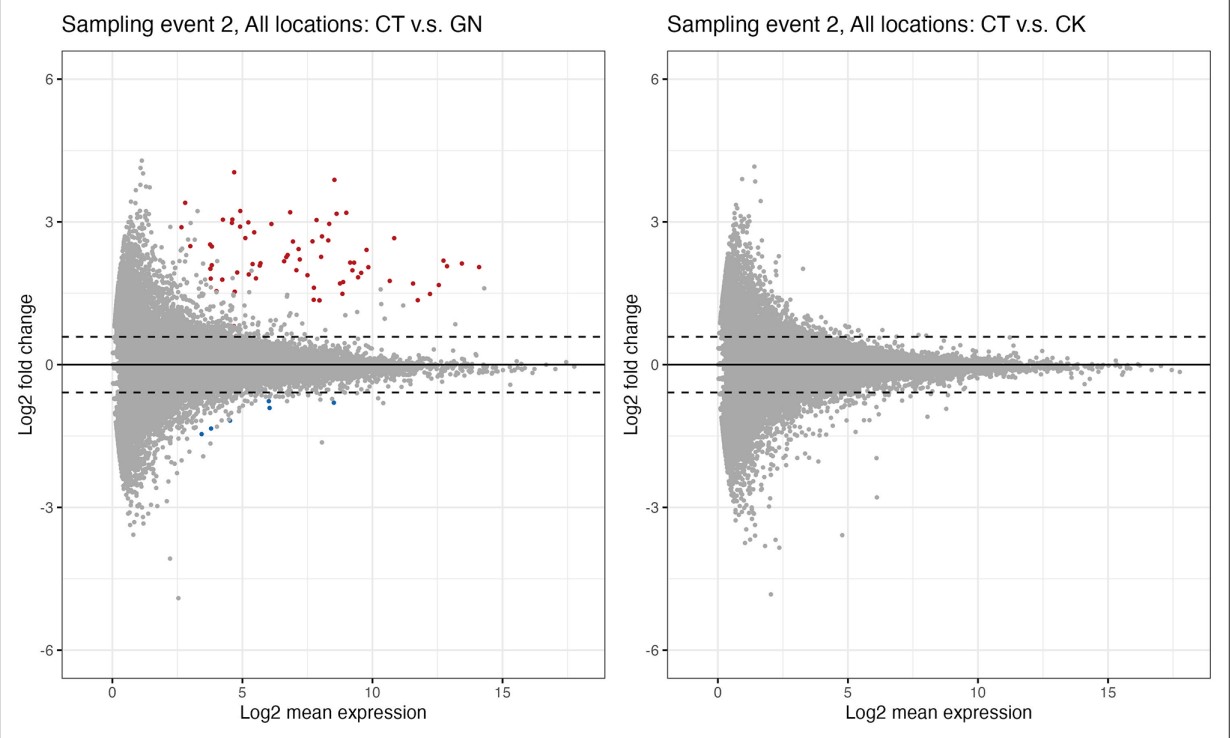

**Figure 5.** Differential expression genes analysis. (**a**)*Globisporangium nunn*-added and (**b**)*Chironomus kiiensis*-removed treatment. Red and blue points indicate significant up- and down-regulated genes, respectively. Upper and lower dashed lines indicate lo$_2$(1.5) and $-$log$_2$(1.5), respectively.

The online version of this article includes the following figure supplement(s) for figure 5:

**Figure supplement 1.** Differential expression gene (DEG) patterns for each Wagner pot (Pot location-specific analysis).

### Changes in rice gene expression pattern in 2019

We further explored changes in the rice performance before and after the manipulation experiments using RNA-seq. We collected rice leaf samples four times (1 day before the first manipulation and 1, 14, and 38 days after the third manipulation). As a result of the RNA-seq analysis, 875,534,703 reads were generated (*Supplementary file 7*), and 8,105,902 reads per sample were used for the sequence analysis. DESeq2 analysis was performed, and here we focus on the results of the second sampling event (i.e. 1 day after the third manipulation). The analysis suggested that, when all three rice pot locations within each plot were merged (see *Figure 3a* for the pot locations), there were differentially expressed genes (DEGs) in the GN treatment (*Figure 5a*; see *Supplementary file 7* for nuclear DEGs) while there were no differentially expressed genes in the CK treatment (*Figure 5b*). However, when the data was separately analyzed for each Wagner pot location, we found that there were several differentially expressed genes even in the CK treatment (*Figure 5—figure supplement 1* and *Supplementary file 8*), suggesting that there was a pot location-specific effect and that the CK treatment also had small effects on the rice gene expression. We found almost no DEGs for leaf samples taken one day before and 14 and 38 days after the third manipulation (the leaf sampling event 1, 3, and 4), suggesting that the influences of the treatments on the gene expression patterns were transient. In *Figure 6*, we show examples of differentially expressed genes. In the GN treatment, the expressions of three genes, Os12g0504050, Os11g0184900, and Os01g0678500 (p<0.0001, negative binomial GLMM; *Figure 6a–c*), were increased, while those of three other genes (Os01g0642200, Os08g0162800, and Os03g0285700) were decreased (p<0.0001, negative binomial GLMM; *Figure 6d–f*). These DEGs are mostly related to the photosynthetic system and stress responses (*Supplementary file 7*).

### Discussion

In the present study, we demonstrated a novel research framework to integrate the ecological network concept and agricultural practices to screen potentially influential organisms. We performed intensive

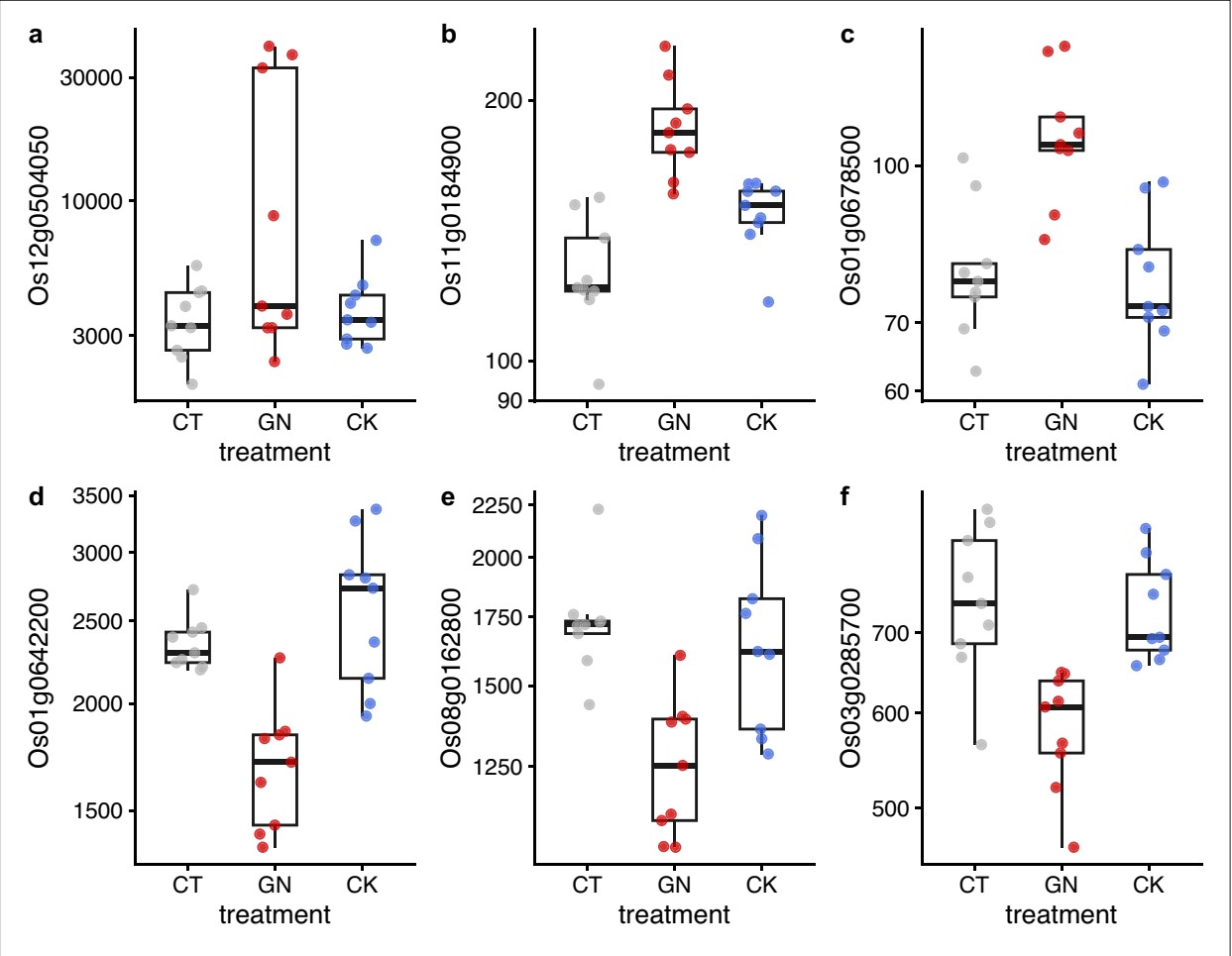

**Figure 6.** Examples of differentially expressed genes after the manipulation experiment. Results of (**a**) Os12g0504050, (**b**) Os01g0642200, (**c**) Os08g0162800, (**d**) Os11g0184900, (**e**) Os03g0285700, and (**f**) Os01g0678500 are presented. *y*-axis represents DESeq2-normalized read counts. Gray, red, and blue indicate CT (control), GN (*Globisporangium nunn* added), and CK (*Chironomus kiiensis* removed) treatments, respectively. The gene expressions of the GN treatment in all six genes are statistically clearly different from those of the other two treatments (p<0.0001) except for the panel GN vs. CK in **c** (p=0.00014) and GN vs. CK in (**d**) (p=0.0087).

field monitoring in 2017, obtained intensive and extensive time series of rice and ecological communities, detected potentially influential organisms using time series-based causality analysis, and based on the results, we performed field manipulation experiments and evaluated the effect of the manipulation. Throughout the study, we detected statistically clear effects of the manipulation on the rice growth and gene expression patterns especially for the GN treatment, proving that our proposed framework might work as a novel approach to comprehensively detect potentially influential, previously overlooked ecological community members for an agricultural system (in our case, rice). Below, we discuss the results, advantages, disadvantages, potential limitations, and future perspectives of our approach.

## Monitoring of ecological community using eDNA and rice growth

Our framework started from monitoring ecological community members using quantitative eDNA metabarcoding (*Ushio, 2022*). We successfully detected more than 1000 ASVs and described detailed temporal dynamics of the ASVs (*Figure 1d and e*; *Ushio, 2022*). We included spike-in standard DNAs whose concentrations were known a priori, and our previous study had shown that, in almost all cases, there were clear positive linear correlations between the sequence reads and the copy numbers of the spike-in standard DNAs (e.g. Supplementary figures in *Ushio, 2022*). This suggested that our quantitative eDNA metabarcoding reasonably quantified the concentrations of eDNA, which can be

a 'rough' index of species abundance in the artificial rice plots. Some time-series-based causal inference methods required quantitative data (i.e. relative abundance data that is typical for amplicon sequencing is not suitable; *Osada et al., 2023*; *Schreiber, 2000*; *Sugihara et al., 2012*), and our method justifies the use of the advanced time series method to infer the causal relationship between rice growth and ecological community members.

Rice growth was monitored by a manual measurement, for example, rice height measurements made using rulers (*Figure 1b*). Although the measurements successfully captured the seasonal dynamics of the rice growth, one might speculate whether only a single variable is sufficient to capture the rice performance. Indeed, it is true that assessing more variables, for example, gene expression (*Nagano et al., 2019*; *Nagano et al., 2012*) and multispectral image monitoring during the whole growth season (*Candiago et al., 2015*), would be potentially useful to fully capture the trajectory of the rice growth. Nonetheless, we suggest that very frequent monitoring of a single variable is sufficient to reconstruct and predict the dynamics of the whole system dynamics (*Takens, 1981*). Thus, we suggest that our monitoring approach can resolve the dynamics at least as a proof-of-concept study, but taking more variables at fine time resolution is certainly beneficial for more accurate delineation of the system dynamics.

## Detection of potentially influential organisms using nonlinear time series analysis

By using UIC (*Osada et al., 2023*), we detected 718 potentially influential ASVs. Short read amplicon sequencing and conservative taxa assignment algorithm of Claident (*Tanabe and Toju, 2013*) did not allow to assign the species name to most of the ASVs, and we could assign the species name only for 52 ASVs. Among them, we detected a previously known detrimental fungus, *Penicillium citrinum*, that may colonize rice and produce a toxin called citrinin that could cause human health problems (*Ali, 2018*; *Ostry et al., 2013*), which partially supported the validity of our time series analysis. However, at least at the time of the analysis, it is unknown whether the species in *Supplementary file 1* really include influential organisms.

In 2019, we focused on two species, *G. nunn* and *C. kiiensis* (*Figure 3—figure supplement 1*). At the time of that analysis, these choices were reasonable: *G. nunn* was previously reported to have some antagonistic activity against pathogens to vegetables such as *Pythium ultimum* (*Kobayashi et al., 2010*) and did not have any clear detrimental effects on rice (which was necessary in order not to spread potential pathogens to nearby farms), and *C. kiiensis* (midge) is a common freshwater insect species in the study region. *C. kiiensis* may have effects on water quality (e.g. the concentration of phosphorus) through their feeding behavior (*Kawai et al., 2003*). However, they have never been focused on in terms of rice growth management, and thus, they provided an opportunity to test the validity of our approach in the present study.

## Field manipulation experiments in 2019

In 2019, we performed the manipulation experiments. Although the CK treatment did not change the community composition statistically clearly, the GN treatment changed the community compositions (*Figure 3b–d*). In the GN treatment, the abundance of putative *Globisporangium* spp. clearly increased (*Figure 3b*), suggesting that the introduction of *G. nunn* was successful. However, we also observed increases in the putative *Globisporangium* spp. abundance in some of the CT and CK samples. The rice plots were located close to each other (50–60 cm apart; *Figure 3a*) and thus immigrations might have occurred between the GN and the other plots. As for *C. kiiensis*, we did not observe a statistically clear decrease in *C. kiiensis* eDNA in the CK treatments (*Figure 3c*), suggesting that *C. kiiensis* individuals or their eDNA persisted even after the removal treatment. This might suggest that the treatment was not successful, or that quantitative eDNA metabarcoding failed to capture the actual decrease in their abundance. On the other hand, we observed statistically clear increases in *C. kiiensis* eDNA in the CT and GN treatments. This response was unexpected, but the effects of the increase in *G. nunn* might be transmitted to other species through the interaction networks which were reconstructed in the previous study (*Ushio, 2022*).

As for the rice growth, although the manipulations were mild (the GN treatment) or might have been relatively unsuccessful (the CK treatment) compared with other conventional manipulations such as the addition of fertilizers and insecticides, we surprisingly found statistically clear effects of the

manipulations under field conditions (*Figure 4b*). The addition of *G. nunn* increased the cumulative rice growth and changed the gene expression pattern after the manipulations statistically clearly. The removal of *C. kiiensis* also affected the rice growth (*Figure 4b*), and the gene expression patterns were changed when the pot locations were taken into account (*Figure 5—figure supplement 1c and f* and *Supplementary file 8*), suggesting that the effects of ecological community members might be situation dependent (e.g., microclimate changes depending on the pot locations). The unclear effects of the CK treatment relative to those of the GN treatment could be due to the relatively unstable removal method (i.e. *C. kiiensis* larvae were manually removed by a hand net) or incomplete removal of the larvae (some larvae might have remained after the removal treatment). Unfortunately, detailed mechanisms of these effects cannot be elucidated in the present study, but some inferences could be made from the gene expression patterns. For example, nuclear DEGs included genes related to photosynthesis and stress responses (*Supplementary files 7 and 8*). These changes were observed relatively quickly and transient (only found in leaf samples collected one day after the third manipulation; *Supplementary file 3*) and thus they could be direct effects from *G. nunn*. Up-regulations of photosynthesis-related genes (e.g. Os08g0433350, Os04g0473150, Os12g0504050, and Os01g0791033 in *Supplementary file 7*) might have contributed to the increase in the cumulative rice heights. Stress response genes such as Os11g0184900 might also have contributed to the changes in the rice performance, but further studies are required to elucidate the mechanisms by which ecological community members affect the rice performance under field conditions.

In the present study, we could not test the effect of other species on rice growth due to the limitations of time, labor, and money, but the species listed in *Supplementary file 1* could be candidates of novel biological agents that influence the rice performance. Thus, validating the effects of these species on the rice performance is important, and targeting ASVs with a large TE value would be a promising direction to find previously overlooked beneficial/detrimental organisms for rice under field conditions. In addition, we could not test the effects of species that had no clear effects on the rice performance according to the time series analysis. It will also be important to validate that such species do not have clear influence on the rice performance in the field, which would further validate the effectiveness of our research framework.

## Potential limitations and future perspectives

Although the present study has shown that our framework is potentially useful for detecting potentially influential, previously overlooked organisms for rice growth under field conditions, there are potential limitations that should be acknowledged.

One limitation is the quantitative capacity and species identification accuracy of the eDNA metabarcoding. As for the first issue, the quantitative eDNA metabarcoding approach used in this study provides reasonable estimates of eDNA concentrations (see the following studies for the comparison between quantitative eDNA metabarcoding and the fluorometric method and quantitative PCR; *Ushio, 2022*; *Ushio et al., 2018b*), and the eDNA concentrations usually positively correlate with the number or biomass of individuals. However, there are species- and environment-specific eDNA dynamics such as degradation and release processes (*Dejean et al., 2011*; *Maruyama et al., 2014*), which may bias the estimation of absolute abundance. Although our causality detection might not be strongly influenced by the species-specific bias (i.e. time series were standardized before the analysis), accurate estimations of the absolute abundance will improve the accuracy of analyses. Identifications and enumerations of individuals using eDNA would be a fascinating direction to improve the accuracy of the species abundance estimations. As for the second issue, short-read sequencing has dominated current eDNA studies, but it is often not sufficient for lower-level taxonomic identification (i.e. species or subspecies identification). Using long-read sequencing techniques (e.g. Oxford Nanopore MinION) for eDNA studies is a promising approach to overcome the second issue (*Baloğlu et al., 2021*).

Another limitation is that the monitoring of rice and the introduced organisms in this study was not comprehensive. While intensive monitoring was conducted during the rice growing season, only a limited number of variables were measured for a limited number of rice individuals and the detailed dynamics of the two introduced species was unclear (i.e. the fate of the introduced species). This is particularly important for understanding how the introduced organisms affected rice performance. At the molecular level, acquiring larger scale 'omics' data for rice and ecological community members, for example long-term gene expression monitoring data (*Nagano et al., 2019*), will

facilitate understanding of detailed mechanisms. In addition, at the field level, future studies should consider applying a more advanced monitoring approach, such as unamended aerial vehicle (*Fujiwara et al., 2022*), multispectral camera (*Candiago et al., 2015*), and/or neural network-based image analysis (*Toda and Okura, 2019*).

Lastly, while we used an advanced nonlinear time series analysis to detect potential causal organisms for rice growth (*Osada et al., 2023*), there is still room for improvement. Nonlinear time series analysis is also a rapidly developing field, and many new techniques are emerging for causal inference (*Runge et al., 2019*; *Suzuki et al., 2022*), quantification of interaction strength (*Chang et al., 2021*) and near future forecasting (*Chen et al., 2020*). Applying these techniques to agricultural data may help to identify previously overlooked beneficial organisms under field conditions.

### Conclusions

In this study, we demonstrated that intensive monitoring of an agricultural system and the application of nonlinear time series analysis are helpful for identifying potentially influential organisms under field conditions. Although the effects of the two species were relatively small and how the two species influenced the rice performance (i.e. the fate of the introduced species) was still unclear, the research framework presented here has future potential. The intensive monitoring, nonlinear time series analysis, and in situ validation of the statistical results may be applicable for other agricultural and aquatic systems. For example, the application of our framework to fishery aquaculture systems may be helpful for detecting potential pathogens of fish. Also, if the continuous monitoring of soil ecological communities becomes possible, our approach can be extended to other crops and vegetables. An automated, real-time monitoring system combined with advanced machine learning methods would provide a promising tool for increasing the accuracy and applicability of our approach. In conclusion, we proposed a novel framework to integrate the ecological network concept and agricultural practices to explore and detect potentially influential organisms. Our proof-of-concept study would be an important basis for the further development of field-based system management.

## Materials and methods

### Field experimental setting and rice monitoring in 2017

Five artificial rice plots were established using small plastic containers (90×90 × 34.5 cm; 216 L total volume; Risu Kogyo, Kagamigahara, Japan) in an experimental field at the Center for Ecological Research, Kyoto University, in Otsu, Japan (34° 58′ 18″ N, 135° 57′ 33″ E) (*Figure 1a* and *Figure 1— figure supplement 1a and b*). Sixteen Wagner pots (φ174.6–160.4 mm [top–bottom] × 197.5 mm; AsOne, Osaka, Japan) were filled with commercial soil, and three rice seedlings (var. Hinohikari) were planted in each pot on 23 May 2017 and then harvested on 22 September 2017 (122 days). The containers ('plots') were filled with well water. This experimental system was previously reported in *Ushio, 2022*, which investigated the mechanism of ecological community dynamics. In the present study, we focused on the relationship between rice performance and ecological community members.

Daily rice growth was monitored by measuring rice leaf height of target individuals every day using a ruler (the highest leaf heights were measured). We selected four rice individuals at the center of each plot as the target individuals (indicated by the four red points in the right panel of *Figure 1a*). The average height of the four individuals were used as a representative rice height of each plot (a time-lapse movie that shows rice growth is available here: https://doi.org/10.6084/m9.figshare.19029650. v1). Leaf SPAD was also measured every day using a SPAD meter (SPAD-502Plus, KONICA-MINOLTA, Inc, Tokyo, Japan) for the same leaf whose height was measured, but not analyzed intensively in the present study. Climate variables (temperature, light intensity, and humidity) were monitored using a portable, automated climate logger (Logbee, Chitose Industries, Osaka, Japan). The Logbee data was corrected and refined using the climate station data provided by the Center for Ecological Research, Kyoto University.

### Water sampling for the eDNA-based monitoring of ecological communities in 2017

The complete information about eDNA-based ecological community monitoring is available in *Ushio, 2022* (https://ndownloader.figstatic.com/files/34067324 and https://ndownloader.figstatic.com/files/

34067327). Approximately 200 ml of water in each rice plot was collected from each of the four corners of the plot using a 500 ml plastic bottle and taken to the laboratory within 30 min. The water was filtered using Sterivex filter cartridges (Merck Millipore, Darmstadt, Germany). Two types of filter cartridges were used to filter water samples: to detect microorganisms, φ0.22-μm Sterivex (SVGV010RS) filter cartridges that included zirconia beads inside were used (*Ushio, 2019*), and to detect macroorganisms, φ0.45-μm Sterivex (SVHV010RS) filter cartridges were used. After filtration, 2 ml of RNAlater solution (ThermoFisher Scientific, Waltham, Massachusetts, USA) were added to each filter cartridge to prevent DNA degradation during storage. In total, 1220 water samples (122 days × 2 filter types × 5 plots) were collected during the census term. In addition, 30 field-level negative controls, 32 PCR-level negative controls with or without the internal standard DNAs, and 10 positive controls to monitor the potential DNA cross-contamination and degradation during the sample storage, transport, DNA extraction and library preparations were used.

## Quantitative analysis of environmental DNA: Library preparations, sequencing, and sequence data processing

Detailed protocols of the quantitative eDNA metabarcoding are described in *Ushio, 2022*, and the protocols below were also adopted for the samples collected in 2019. Briefly, DNA was extracted and purified using a DNeasy Blood & Tissue kit (Qiagen, Hilden, Germany). After the purification, DNA was eluted using 100 μl of the elution buffer and stored at −20 °C until further processing.

A two-step PCR approach was adopted for the library preparation for quantitative MiSeq sequencing. The first-round PCR (first PCR) was carried out with the internal standard DNAs to amplify metabarcoding regions using primers specific to prokaryotes (515 F and 806 R; *Bates et al., 2011*; *Caporaso et al., 2011*), eukaryotes (Euk_1391 f and EukBr; *Stoeck et al., 2010*), fungi (ITS1-F-KYO1 and ITS2-KYO2; *Toju et al., 2012*) and animals (mlCOIintF and HCO2198' *Folmer et al., 1994*; *Leray et al., 2013*). The second-round PCR (second PCR) was carried out to append indices for different samples for sequencing with MiSeq. The DNA library was sequenced on the MiSeq (Illumina, San Diego, CA, USA).

The raw MiSeq data were converted into FASTQ files using the bcl2fastq program provided by Illumina (bcl2fastq v2.18). The FASTQ files were then demultiplexed using the command implemented in Claident v0.2.2019.05.10 (http://www.claident.org; *Tanabe and Toju, 2013*). Demultiplexed FASTQ files were then analyzed using the Amplicon Sequence Variant (ASV) method implemented in 'dada2' package (*Callahan et al., 2016*) of R (*R Development Core Team, 2022*). Taxonomic identification was performed using Claident (*Tanabe and Toju, 2013*). We detected negligible sequences from the negative control samples.

## Estimations of DNA copy numbers and validation of the quantitative capability of the MiSeq sequencing with internal standard DNAs

For all analyses after this subsection, the free statistical environment R was used (*R Development Core Team, 2022*). The procedure used to estimate DNA copy numbers consisted of two parts, following previous studies (*Ushio, 2019*; *Ushio et al., 2018b*). Briefly, we performed (i) linear regression analysis to examine the relationship between sequence reads and the copy numbers of the internal standard DNAs for each sample, and (ii) conversion of sequence reads of non-standard DNAs to estimate the copy numbers using the result of the linear regression for each sample. The regression equation was: MiSeq sequence reads = sample-specific regression slope ×the number of standard DNA copies [/μl]. Then, the estimated copy numbers per μl extracted DNA (copies/μl) were converted to DNA copy numbers per ml water in the rice plot (copies/ml water). The quantitative capacity of this method was thoroughly evaluated by comparing with quantitative PCR, fluorescence-based DNA measurement, and shotgun metagenomic analysis (*Ushio, 2022*; *Ushio et al., 2018b*), and the method was shown to have reasonable capacity to quantify DNA. The method has already been shown to be effective for eDNA-based quantitative assessment of ecological community (*Sato et al., 2021*; *Tsuji et al., 2020*). The estimation of eDNA concentrations was necessary to apply the time series-based causality analysis explained in the next section.

## Detection of potential causal relationships between rice growth and ecological community members

We detected potentially influential organisms for rice growth analyzing the quantitative, 1197-species eDNA time-series (*Ushio, 2022*) and the daily rice growth rate obtained in 2017 using nonlinear time

series analysis. We quantified information flow from eDNA time series to rice growth rate (i.e. a proxy of interaction between an organism and rice growth) by the 'unified information-theoretic causality (UIC)' method (*Osada et al., 2023*) implemented in the 'rUIC' package (*Osada and Ushio, 2021*) of R. UIC tests the statistical clarity of information flow between variables in terms of transfer entropy (TE; *Frenzel and Pompe, 2007*; *Schreiber, 2000*) computed by nearest neighbor regression based on time-embedded explanatory variables (i.e. cross-mapping: *Sugihara et al., 2012*). In contrast to the standard method used to measure TE, UIC quantifies information flow as follows:

$$TE = \frac{1}{T} \sum_{t=1}^{T} log \left( \frac{p \left( y_{t+tp} \mid x_t, x_{t-\tau}, \cdots, x_{t-(E-1)\tau}, z_t \right)}{p \left( y_{t+tp} \mid x_{t-\tau}, x_{t-2\tau}, \cdots, x_{t-(E-1)\tau}, z_t \right)} \right),$$ (1)

where $x$, $y$, and $z$ represent an effect variable, a potential causal variable, and a conditional variable (in our case, $z$ is/are climate variable(s) if they are causal factors), respectively. $p\left(A \mid B\right)$ represents conditional probability: the probability of $A$ conditioned on $B$. $t$, $tp$, $\tau$, and $E$ represent the time index, time step, a unit of time lag, and the optimal embedding dimension, respectively. $T$ is the number of times to evaluate TE. For example, if $tp = -1$ in Eqn. (1), UIC tests the causal effect from $y_{t-1}$ to $x_t$. Optimal $E$ was selected by measuring TE as follows:

$$TE = \frac{1}{T} \sum_{t=1}^{T} log \left( \frac{p \left( x_{t+tp} \mid y_t, x_t, x_{t-\tau}, \cdots, x_{t-(E-1)\tau}, z_t \right)}{p \left( x_{t+tp} \mid y_t, x_t, x_{t-\tau}, \cdots, x_{t-(E_R-1)\tau}, z_t \right)} \right),$$ (2)

where $E_R\left(E\right)$ is the optimal embedding dimension of lower dimensional models. Eqn. (2) is a TE version of simplex projection (*Sugihara and May, 1990*), and we set tp = 1 in Eqn. (2) to determines the optimal $E$ (i.e. the optimal $E$ based on one-step forward prediction). Statistical clarity was tested by bootstrapping data after embedding (a threshold was set to 0.05). TE measured according to Eqn. (1) gains the advantage of previous causality tests, that is, standard TE methods (*Runge et al., 2012*; *Schreiber, 2000*) and convergent cross mapping (CCM) (*Sugihara et al., 2012*), as explained in *Osada et al., 2023*.

By using UIC, we quantified TE from eDNA time series to rice growth. We standardized the eDNA time series (copies/ml water) and rice growth rates (cm/day) to have zero means and a unit of variance before the analysis. We tested time-lag up to 14 days (i.e. $tp$ in Eqn. (1) was from 0 to –14; effects from up to 14 days ago were considered). We interpret a finding that TE from eDNA time series to rice growth rate is statistically greater than zero to mean that the ecological community member statistically clearly influences rice growth rate.

## Field experimental setting in 2019

Based on the results in 2017, we performed a field manipulative experiment. We focused on two potentially influential organisms as explained in the next section. Nine artificial rice plots were established using smaller plastic containers than those used in 2017 (42.2 × 32.0 × 30.0 cm; 40.5 L total volume; Sanko, Tokyo, Japan) in the same experimental field at Kyoto University (i.e. the identical location with Plot 3 in 2017) (*Figure 3a*). Three Wagner pots were filled with commercial soil, and three rice seedlings (var. Hinohikari) were planted in each pot on 20 May 2019. Three treatments were prepared (see the next section), including the control treatment, and there were three replicates for each treatment (3 treatments × 3 replicates × 3 Wanger pots × 3 rice seedlings = 81 rice seedlings in nine rice plots). The rice individuals were harvested on 20 September 2019 (124 day growing period).

## Field manipulation experiments in 2019

Field manipulation experiments were performed using two potentially influential species, *Globisporangium nunn* (Oomycete; previously known as *Pythium nunn*; *Uzuhashi et al., 2010*) and *Chironomus kiiensis* (midge species; *Figure 3—figure supplement 1*). The reasons why we focused on these two species are described in the Results. As for *G. nunn*, we referred to the ASV found by the eDNA metabarcoding in 2017 as 'putative *Globisporangium*' because there are differences in the ASV sequence and the *G. nunn* type strain sequence. Detailed explanations about the difference in the sequences are explained in 'Notes on *Globiosporangium nunn*' in *Supplementary file 2*.

We performed the rice plot manipulations three times at around noon on 24, 26, and 28 June 2019. There were three treatments, namely, 'Globisporangium nunn-added' (GN), 'C. kiiensis-removed' (CK), and control (CT) treatments with three replicates for each treatment. For the GN treatment, isolated and incubated G. nunn was mixed in vermiculite (**Kobayashi et al., 2010**; **Tojo et al., 1993**) and approximately 200 ml of vermiculite was added to each Wagner pot (**Figure 3—figure supplement 1a and b**). Vermiculite without G. nunn was added to Wagner pots in the two other treatments. For the CK treatment, midge larvae were removed instead of adding them because isolation and cultivation of this insect species are technically difficult and time- and effort-consuming. Midge larvae were removed using a φ1.0 mm net (**Figure 3—figure supplement 1c**). On average, 283 midge larvae were removed from each plot and the DNA of about 100 removed larvae was sequenced using Sanger sequencing, and all were confirmed as C. kiiensis. For the other treatments, the net was also inserted into plots, the water in each plot was mixed by the net for a while, but midge larvae were not removed.

## Rice growth and ecological community monitoring, and rice leaf sampling during the manipulation experiments

Rice growth was monitored by measuring rice leaf height of target individuals using a ruler, as in 2017. We selected one rice individual in the middle of each of the Wagner pots as the target individuals (indicated by the three red points in the inlet panel of **Figure 3a**; labelled as locations '1', '2', and '3' from east to west). Leaf SPAD was also measured using a SPAD meter as in 2017. The ecological communities in the rice plots were monitored by analyzing eDNA as in 2017. After we harvested rice, we counted the number of fertile and sterile grains and quantified rice yields.

In 2019, the rice growth was monitored weekly or biweekly except during the period of the manipulation experiments. The manipulations were performed three times (i.e. 24, 26, and 28 June 2019), and the daily monitoring of the rice plots was performed before and after the manipulations (**Supplementary file 3**). Rice performance (growth rates and SPAD) was monitored every day from 11 June to 12 July (32 days). Water samples for eDNA-based ecological community monitoring were collected every day from 18 June to 12 July (25 days). Rice leaf samples for RNA-seq were collected four times before and after the field manipulations (**Supplementary file 3**). The leaf samples were immediately frozen by dipping into liquid nitrogen under field conditions, and kept frozen until stored in a freezer at –20 °C. In total, 108 leaf samples were collected for RNA-seq analysis (**Supplementary file 7**).

## Effects of the field manipulation experiment on rice growth and ecological communities

The differences in rice performance among the three treatments and total eDNA concentrations of ecological community members (copies/ml water) before and after the manipulation experiments were tested using linear mixed model (LMM) and general linear mixed model (GLMM), respectively, using 'lme4' package of R (**Bates et al., 2015**). For rice growth rates, the effect of the treatment was tested with the random effects of rice individual and rice plot (in R, this is `lme4::lmer(growth_rate ~treatment + (1|ind/plot), data = data)`), and the effects were separately analyzed for before/after the manipulation. Rice cumulative growth (i.e. cumulative growth before the third manipulation or 10 days after the third manipulation) was analyzed similarly, but the random effect of rice individuals was not included because there was only one cumulative value for each rice individual. Also, we tested alternative models including the timing and manipulation treatments (in R, these are `lme4::lmer(growth_rate ~before_or_after_manipulation*treatment + (1|ind/plot), data = data)` and `lme4::lmer(cumulative_growth ~before_or_after_manipulation*treatment + (1|plot), data = data)`), and obtained general agreement with the results of the above-mentioned analysis. For ecological community, we analyzed the eDNA data taken during the intensive monitoring period before and after the field manipulation experiments (18 June to 12 July; **Supplementary file 3**), and the effects of the manipulation (i.e. the addition of G. nunn or the removal of C. kiiensis) were separately tested for each treatment with the random effect of rice plot assuming a gamma error distribution (in R, this is `lme4::glmer(DNA_conc ~ before_or_after_manipulation + (1|plot), data = data, family = Gamma(link="log"))`). Also, as in the rice growth analysis, we tested alternative models including the timing and manipulation treatments (in R, these are `lme4::glmer(DNA_conc ~ before_or_after_manipulation*treatment +`

(1|plot), data = data, family = Gamma(link="log"))). Differences in the ecological community compositions were visualized using t-distributed stochastic neighbor embedding (t-SNE) (*Van der Maaten and Hinton, 2008*). In the present study, we used the term 'statistical clarity' instead of 'statistical significance' to avoid misinterpretations, according to the recommendations by *Dushoff et al., 2019*, when p<0.05.

## Quantifications of RNA expressions of rice leaves

The leaf samples were ground under cryogenic conditions using a Multi-Beads Shocker (Yasui Kikai, Osaka, Japan). Total RNA was extracted using the Maxwell 16 LEV Plant RNA Kit (Promega, Madison, WI, USA). RNA concentration was measured using the broad-range Quant-iT RNA Assay Kit (Thermo Fisher Scientific, Waltham, MA, USA). The RNA concentrations were adjusted and 500 ng of RNA was used as the input of each sample for library preparation. Library preparation for RNA-sequencing was conducted using Lasy-Seq (*Kamitani et al., 2019*) version 1.1 (https://sites.google.com/view/lasy-seq/). The library was sequenced using HiSeq X (Illumina, San Diego, CA, USA) with paired-end sequencing lengths of 150 bp. On average, 8,105,902 reads per sample (±5,494,529 S.D.) were generated (total reads = 875,534,703 reads; *Supplementary file 7*).

All obtained reads were trimmed using Trimmomatic version 0.33 (*Bolger et al., 2014*) using the following parameters: TOPHRED33, ILLUMINACLIP:TruSeq3-SE.fa:2:30:10, LEADING:19, TRAILING:19, SLIDINGWINDOW:30:20, AVGQUAL:20, MINLEN:40, indicating that reads with more than 39 nucleotides and average quality scores over 19 were reported. Then, the trimmed reads were mapped onto the reference sequences of the IRGSP-1.0_transcript (*Kawahara et al., 2013*) and the virus reference sequences, which were composed of complete genome sequences of 7,457 viruses obtained from NCBI GenBank (*Kashima et al., 2021*) using RSEM version 1.3.0 (*Li and Dewey, 2011*) and Bowtie version 1.1.2 (*Langmead et al., 2009*) with default parameters. The output of the analysis, 'expected_counts', was used as inputs to the analysis of differentially expressed genes.

## Detection of differentially expressed genes (DEGs)

The expected counts were imported as a phyloseq object using 'phyloseq' package (*McMurdie and Holmes, 2013*) of R (*R Development Core Team, 2022*). Then, the object was converted to a DESeq2 object using `phyloseq::phyloseq_to_deseq2()` function, and differentially expressed genes (DEGs) were detected using 'DESeq2'" package (*Love et al., 2014*) of R. DESeq2 provides statistical frameworks for determining differential expression using a model based on the negative binomial distribution. Briefly, size factors and dispersions were estimated, generalized linear models (GLMs) assuming a negative binomial error distribution were fitted, and the differences were tested. These analyses were performed using `DESeq2::DESeq()` function implemented in the package. In the DESeq2 analysis, false discovery rate was set as 0.05, and genes with an adjusted p<0.05 found by DESeq2 were assigned as differentially expressed. We measured rice gene expression four times and we detected almost no DEGs at sampling events 1, 3, and 4 except for two DEGs at sampling event 4. Therefore, we report only the results of sampling event 2 in the present study (one day after the third field manipulation). In *Supplementary files 8 and 9*, we report a list of nuclear DEGs.

## Acknowledgements

We thank Asako Kawai and Mutsumi Kato for assistance in field monitoring and DNA library preparations, Yutaka Osada for advice on the data analysis, Akira Matsumoto and Satomi Yoshinami for maintenance of the experimental field and assistance in the field monitoring, Eiso Inoue for comments on the midge species, and Natsumi Fujii for assistance in the field experiment. This research was supported by PRESTO (JPMJPR16O2) from the Japan Science and Technology Agency (JST), KAKENHI (B) 20H03323, the Hakubi Project in Kyoto University, and The Hong Kong University of Science and Technology Startup Fund to MU.

## Additional information

### Funding

| Funder | Grant reference number | Author |
|---|---|---|
| Japan Science and Technology Agency | JPMJPR16O2 | Masayuki Ushio |
| Japan Society for the Promotion of Science | KAKENHI (B) 20H03323 | Masayuki Ushio |
| The Hong Kong University of Science and Technology | Startup Fund | Masayuki Ushio |
| Kyoto University | Hakubi Project | Masayuki Ushio |

The funders had no role in study design, data collection and interpretation, or the decision to submit the work for publication.

### Author contributions

Masayuki Ushio, Conceptualization, Data curation, Formal analysis, Supervision, Funding acquisition, Validation, Investigation, Visualization, Methodology, Writing – original draft, Project administration, Writing – review and editing; Hiroki Saito, Investigation, Methodology, Writing – review and editing; Motoaki Tojo, Resources, Investigation, Methodology, Writing – review and editing; Atsushi J Nagano, Software, Investigation, Methodology, Writing – review and editing

### Author ORCIDs

Masayuki Ushio https://orcid.org/0000-0003-4831-7181
Motoaki Tojo https://orcid.org/0000-0001-9660-4359
Atsushi J Nagano https://orcid.org/0000-0001-7891-5049

Reviewer #1 (Public Review): https://doi.org/10.7554/eLife.87202.3.sa1
Reviewer #2 (Public Review): https://doi.org/10.7554/eLife.87202.3.sa2
Author response https://doi.org/10.7554/eLife.87202.3.sa3

# Additional files

### Supplementary files

Supplementary file 1. Potential causal species for the rice growth.

Supplementary file 2. Notes on *Globiosporangium nunn.*

Supplementary file 3. Field monitoring schedule in 2019.

Supplementary file 4. Effects of the field manipulations on eDNA concentrations of target species using an alternative model.

Supplementary file 5. Effects of the field manipulations on the rice growth using an alternative model.

Supplementary file 6. Rice yield data.

Supplementary file 7. Meta-data for rice leaf samples for RNA expression analysis.

Supplementary file 8. List of differentially expressed genes in the GN treatment.

Supplementary file 9. List of location-specific differentially expressed genes.

MDAR checklist

### Data availability

Computer codes and data used in the study are available in Github (https://github.com/ong8181/rice-ecolnet-2019, copy archived at *Ushio, 2023*) and archived in Zenodo (https://doi.org/10.5281/zenodo.8124471). Sequence data were deposited in DDBJ Sequence Read Archives (DRA). The accession numbers are as follows: DRA009658, DRA009659, DRA009660 and DRA009661 for eDNA data of ecological communities in 2017 (*Ushio, 2022*), DRA015682, DRA015683, DRA015685, and

DRA015686 for eDNA for eDNA data of ecological communities in 2019, and DRA015706 for rice RNA expression data.

The following datasets were generated:

| Author(s) | Year | Dataset title | Dataset URL | Database and Identifier |
|---|---|---|---|---|
| Ushio M | 2020 | eDNA data of ecological communities in 2017 | https://ddbj.nig.ac.jp/resource/sra-submission/DRA009658 | DDBJ Sequence Read Archive, DRA009658 |
| Ushio M | 2020 | eDNA data of ecological communities in 2017 | https://ddbj.nig.ac.jp/resource/sra-submission/DRA009659 | DDBJ Sequence Read Archive, DRA009659 |
| Ushio M | 2020 | eDNA data of ecological communities in 2017 | https://ddbj.nig.ac.jp/resource/sra-submission/DRA009660 | DDBJ Sequence Read Archive, DRA009660 |
| Ushio M | 2020 | eDNA data of ecological communities in 2017 | https://ddbj.nig.ac.jp/resource/sra-submission/DRA009661 | DDBJ Sequence Read Archive, DRA009661 |
| Ushio M | 2023 | eDNA data of ecological communities in 2019 | https://ddbj.nig.ac.jp/resource/sra-submission/DRA015682 | DDBJ Sequence Read Archive, DRA015682 |
| Ushio M | 2023 | eDNA data of ecological communities in 2019 | https://ddbj.nig.ac.jp/resource/sra-submission/DRA015683 | DDBJ Sequence Read Archive, DRA015683 |
| Ushio M | 2023 | eDNA data of ecological communities in 2019 | https://ddbj.nig.ac.jp/resource/sra-submission/DRA015685 | DDBJ Sequence Read Archive, DRA015685 |
| Ushio M | 2023 | eDNA data of ecological communities in 2019 | https://ddbj.nig.ac.jp/resource/sra-submission/DRA015686 | DDBJ Sequence Read Archive, DRA015686 |
| Ushio M | 2023 | Rice RNA expression data | https://ddbj.nig.ac.jp/resource/sra-submission/DRA015706 | DDBJ Sequence Read Archive, DRA015706 |
| Ushio M | 2023 | ong8181/rice-ecolnet-2019: v0.9.0 | https://doi.org/10.5281/zenodo.8124472 | Zenodo, 10.5281/zenodo.8124472 |

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
