## [Editor Report · eLife assessment]

There is a tremendous need to increase agricultural productivity with means that are both practical and efficient. Drawing on data from variable field environments, this **important** study provides a theoretical framework for the identification of new factors with presumed relevance for crop growth. This framework can be applied in the context of both agricultural and ecological studies. There is **solid** evidence for several of the authors' claims, but the impact of the study is limited due to missing functional validation of candidate species in the field. Plant biologists and ecologists working in agricultural and natural environments will find the work interesting.

---

## [Referee Report · Reviewer #1 (Public Review)]

This manuscript describes identification of influential organisms on rice growth and an attempt of validation. The analysis of eDNA on rice pot and mimic field provides rice growth promoting organisms. This approach is novel for plant ecology field. However current results did not fully support whether eDNA analysis-based detection of influencing organism.

The strength of this manuscript is to attempt application of eDNA analysis-based plant growth differentiation. The weakness is too preliminary data and experimental set-up to make any conclusion. The trials of authors experiments are ideal. However, the process of data analysis did not meet certain level. For example, eDNA analysis of different time points on rice growth stages resulted two influential organisms for rice growth. Then they cultivate two species and applied rice seedlings. Without understanding of fitness and robustness, how we can know the effect of the two species on rice growth.

The authors did not check the fate of two species after introducing into rice. If this is true, it is difficult to link between the rice gene expression after treatments and the effectiveness of two species. I think the validation experiment in 2019 needs to be re-conducted.

As authorized gave answered, no strong rationale to select the two species was found. However, I insist that the method has enough novelty to present to general audiences.

---

## [Referee Report · Reviewer #2 (Public Review)]

Most farming is done by subtracting or adding what people want based in nature. However, in nature, crops interact with various objects, and mostly we are unaware of their effects. In order to increase agricultural productivity, finding useful objects is very important. However, in an uncontrolled environment, it coexists with so many biological objects that it is very inefficient to verify them all experimentally. It is therefore necessary to develop an effective screening method to identify external environmental factors that can increase crop productivity. This study identified factors presumed to be important to crop growth based on metabarcoding analysis, field sampling, and non-linear analysis/information theory, and conducted a mesocosm experiment to verify them experimentally. In conclusion, the object proposed by the author did not increase rice yield, but rather rice growth rate.

The authors responded to my general concerns and all of my specific comments. The manuscript has significantly improved. The flow of aims and approaches is more understandable. Extra supplementary material -especially the visual ones, is useful.

I agree with the other reviewers that the study needs more data and evidence. However, this study aims to introduce ecological concepts and advanced statistical methods to the field. Also, most time series analyses require absolute abundance data, but the manuscript provides solutions for the sequencing data.

---

## [Author Response]

The following is the authors’ response to the original reviews.

**eLife assessment**
It is very important to find practical and efficient means in order to increase agricultural productivity. Drawing on data from variable field environments, this study provides a useful theoretical framework to identify new factors that could increase agricultural production. There is solid evidence to support the authors' claims, though following the fate of candidate species after introduction into rice fields would have strengthened the study. Plant biologists and ecologists working in nature and fields will find the work interesting.

Thank you so much for your careful evaluation of our manuscript. We are very pleased to hear that you found our framework useful. We have revised our manuscript according to the "Recommendations for the Authors" to improve our manuscript.

Public Review
**Reviewer #1 (Public Review):**
This manuscript describes the identification of influential organisms on rice growth and an attempt of validation. The analysis of eDNA on rice pot and mimic field provides rice growth promoting organisms. This approach is novel for plant ecology field. However current results did not fully support whether eDNA analysis-based detection of influencing organism.

Thank you so much for evaluating our manuscript. We have carefully read and responded to your comments. We hope our responses resolve your concerns on our study.

The strength of this manuscript is to attempt application of eDNA analysis-based plant growth differentiation. The weakness is too preliminary data and experimental set-up to make any conclusion. The trials of authors experiments are ideal. However, the process of data analysis did not meet certain levels. For example, eDNA analysis of different time points on rice growth stages resulted in two influential organisms for rice growth. Then they cultivate two species and applied rice seedlings. Without understanding of fitness and robustness, how we can know the effect of the two species on rice growth.

We agree with your comments that we did not have the fitness data of the two species and/or rice seedlings. Thus, it is still difficult to obtain deep understanding of the mechanisms of our findings that the species introduced in the system would influence rice growth. Nonetheless, our study demonstrated the effectiveness of our research framework as we found evidence that the species that were discovered by the eDNA monitoring and time series analysis indeed cause changes in the system. We believe that the first step is to show that the framework is workable and that detailed understanding of the mechanisms or genetic pathway was not a focus of our study. To avoid misunderstanding, we have added several explanations regarding this point in L426–431 and L447. For example, in L426, we have added the following statement: "... the detailed dynamics of the two introduced species was unclear (i.e., the fate of the introduced species). This is particularly important for understanding how the introduced organisms affected rice performance...".

The authors did not check the fate of two species after introducing into rice. If this is true, it is difficult to link between the rice gene expression after treatments and the effectiveness of two species. I think the validation experiment in 2019 needs to be re-conducted.

We did not check the fate of the two species (except measuring the eDNA concentrations of the species), and it is true that we cannot show evidence of "how" these two species influence the rice gene expression. Understanding molecular mechanisms of the phenomenon that we found is important (especially from the viewpoint of molecular biology), but our primary objective was to demonstrate that our "eDNA x time series analysis" framework is feasible for detecting previously overlooked but influential organisms. To this end, we believe that we achieved our objective and repeating the validation experiment should be for a different purpose (i.e., for understanding molecular mechanisms). We have clarified these points in L426–431 and L447 as explained above.

**Reviewer #2 (Public Review):**
The manuscript "Detecting and validating influential organisms for rice growth: An ecological network approach" explores the influence of biotic and abiotic entities that are often neglected on rice growth. The study has a straightforward experimental design, and well thought hypothesis for explorations. Monitoring data is collected to infer relationships between species and the environment empirically. It is analyzed with an up-to-date statistical method. This allowed the manuscript to hypothesize and test the effects most influential entities in a controlled experiment.

Thank you so much for your careful evaluations. We are pleased to see that you evaluated our manuscript positively. We have further revised our manuscript according to your comments and hope the revision has resolved your concerns.

The manuscript is interesting and sets up a nice framework for future studies. In general, the manuscript can be improved significantly, when this workflow is smoothly connected and communicated how they follow each other more than the sequence and dates provided. It is valuable philosophical thinking, and the research community can benefit from this framework.

Thank you for your suggestions. In order to improve the logic flow and readability of our manuscript, we have revised the descriptions of workflow and clarified how the experimental and statistical steps were connected to each other. To do so, we have added brief explanations about what/how we did at the first sentence of Results subsections (some of these explanations were only in Materials and Methods in the original manuscript). Also, we have moved all of the Supplementary Materials and Methods to the main text. We have thoroughly revised the manuscript, and we hope that all the parts of our manuscript have been connected more smoothly than in the original manuscript.

I understand the length and format of the manuscript make it difficult to add more details, but I am sure it can refer to/clear some concepts/methods that might be new for the audience. How/why variables are selected as important parts of the system, a tiny bit of information about the nonlinear time series analysis in the early manuscript, and the biological reasoning behind these statistically driven decisions are some examples.

We have explained how/why variables are selected (in L125), added more information about the nonlinear time series analysis (in L129 and L175) , and added the biological reasoning behind the statistical decisions (L195).

**Reviewer #3 (Public Review):**
Most farming is done by subtracting or adding what people want based in nature. However, in nature, crops interact with various objects, and mostly we are unaware of their effects. In order to increase agricultural productivity, finding useful objects is very important. However, in an uncontrolled environment, it coexists with so many biological objects that it is very inefficient to verify them all experimentally. It is therefore necessary to develop an effective screening method to identify external environmental factors that can increase crop productivity. This study identified factors presumed to be important to crop growth based on metabarcoding analysis, field sampling, and non-linear analysis/information theory, and conducted a mesocosm experiment to verify them experimentally. In conclusion, the object proposed by the author did not increase rice yield, but rather rice growth rate.

Thank you so much for your evaluation of our manuscript. We have revised our manuscript based on your comments, and hope it has been improved compared with the original version.

StrengthIn actual field data, since many variables are involved in a specific phenomenon, it is necessary to effectively eliminate false positives. Based on the metabarcoding technique, various variables that may affect rice growth were quantitatively measured, although not perfectly, and the causal relationship between these variables and rice growth was analyzed by using information transfer analysis. Using this method, two new players capable of manipulating rice growth were verified, despite their unknown functions until now. I found this process to be very logical, and I think it will be valuable in subsequent ecological studies.

We are very pleased to see that you found our framework is very logical and potentially beneficial for future ecological studies.

WeaknessesCK treatment's effectiveness remains questionable. Rice's growth was clearly altered by CK treatment. The validation of the CK treatment itself is not clear compared to the GN treatment, and the transcriptome data analysis results do not show that DEG is not present. The possibility of a side effect caused by a variable that the author cannot control remains a possibility in this case. Even though this part is mentioned in Discussion, it is necessary to discuss various possibilities in more detail.

We agree that the effectiveness of the CK treatment was questionable. We have added some more discussion about this point in L376: "The unclear effects of the CK treatment relative to those of the GN treatment could be due to the relatively unstable removal method (i.e., C. kiiensis larvae were manually removed by a hand net) or incomplete removal of the larvae (some larvae might have remained after the removal treatment)."

**Reviewer #1 (Recommendations For The Authors):**
Comment #1-1 This manuscript describes identification of influential organisms on rice growth and an attempt of validation. The analysis of eDNA on rice pot and mimic field provides rice growth promoting organisms. This approach is novel for plant ecology field. However current results did not fully support whether eDNA analysis-based detection of influencing organism.

Thank you for your careful evaluations of our manuscript. We are pleased to see you found that our approach is novel. We have revised our manuscript in accordance with your comments, and we hope that the revision and responses resolved your concerns.

Comment #1-2 1. Experimental setting: Authors made up small scale pot system in 2017 and then expanded manipulative experiment. I do not understand how two influencing organism sequences were identified from the single treatment depending on different time points. How they can be convince the two organisms affect the rice growth rather than other biological and environmental factors.

In 2017, we performed an intensive monitoring of the experimental rice plots and obtained large time series data (122-day consecutive monitoring x 5 plots = 610 data points). The time series data were analyzed using the information-theoretic causal analysis. The analysis is critically different from correlational analyses and designed to identify causal relationships among variables. Although we understand that field manipulation experiments are a common and straightforward approach to identify causal relationships among organisms, we chose the "fieldmonitoring + time-series-based causal analysis" approach. This is because, as explained in the main text, there are numerous factors that could influence rice performance, and it is practically impossible to perform manipulative experiments for all the potential factors that could influence rice growth. On the other hand, our "field-monitoring + timeseries-based causal analysis" approach has a potential to identify multiple factors under field conditions, even by the single experimental treatment.

Nonetheless, we must admit that our time-series-based approach still has a chance to misidentify causal factors. Our framework relies on statistics, so the chance of false-positive detection of causality cannot be zero. This was exactly the reason why we performed the "validation" experiment in 2019. To complement the statistical results of the 2017 experiments, we performed another experiment in 2019.

Comment #1-3 2. eDNA technology: The eDNA analysis based on four universal primers 16s rRNA, 18s rRNA, ITS, and COI regions must not be enough to identify specific species. The resolution of species classification may not meet to confirm exact species. Thus, the accuracy of two species that they selected for further experiment is difficult to be confirmed. Authors also referred to "putative Globisporangium".

Your point is correct. The DNA barcoding regions we selected are short and it is often difficult to identify species. However, this limitation could not have been overcome even if we had chosen a different genetic marker. The long-read sequencing technology could partially solve the issue, but the number of sequence reads generated by the long-read technique is less than that by the short-read sequencing technology, and comprehensive detection of all species in an ecological community was still challenging. Our approach struck a balance among the identification resolution, comprehensiveness of the analysis, and sequencing costs. In addition, even though we could not identify most ASVs at the species level, some ASVs could be identified at the species level (52 ASVs among the 718 ASVs which had causal influences on rice growth), and we selected the two species (G. nunn and C. kiiensis) from the 52 species.

Further, the taxa assign algorithm we used here (i.e., Claident; Tanabe & Toju 2012 PLoS ONE 10.1371/journal.pone.0076910) adopted conservative criteria for species identification and has a low falsepositive probability.

More importantly, this is also the reason why we performed the "validation" experiment in 2019. The species identified in the 2017 experiment are still "potential" organisms that influence rice growth (i.e., the hypothesis-generating phase), and we tested the hypothesis in 2019.

Nonetheless, we must admit that clear description of potential limitations is important. Thus, we have discussed this in L418: "As for the second issue, short-read sequencing has dominated current eDNA studies, but it is often not sufficient for lower-level taxonomic identification. Using long-read sequencing techniques (e.g., Oxford Nanopore MinION) for eDNA studies is a promising approach to overcome the second issue".

Comment #1-4 3. Biological relevance 1: Authors identify two organisms as influencing organism for rice growth. As conducting the first experiment in 2017, the 2019 experiment was different from natural condition. The two experiments in 2017 and 2019 were conducted under different conditions. How do they compare the experiments? At least, the eDNA analyses in 2017 and 2019 should be very similar. I cannot find such data.

The experimental conditions were different between 2017 and 2019 because they were conducted in different years. Theoretically, it is ideal if the experimental conditions in 2019 are covered by the range of experimental conditions in 2017 (e.g,. rice variety, air temperature, rainfall, and solar radiation). If this condition were satisfied, the attractor (i.e., rice growth trajectory delineated in the state space) in 2019 would be within that in 2017, and our model prediction in 2017 would be used to predict dynamics in 2019 accurately. To fulfill the conditions, we made as much effort as possible: we used the same rice variety and soils in 2019 as those used in 2017, and started our experiment at the same timing in 2019 as that in 2017.

Although natural ecological dynamics cannot be precisely controlled, our monitoring revealed that the ecological dynamics in 2019 was qualitatively similar to that in 2017. To demonstrate that the experimental conditions and eDNA community data were similar between the two experiments, we have presented the climate and eDNA data in an inset figure in Figure 3a, Figure 1–figure supplement 2, Figure 3–figure supplement 2. We must admit that these dynamics are not identical, but we hope that this resolves your concern.

Comment #1-5 4. Lack of detail description: In the Materials and Methods, there are many parts which lack on detail description. For instance, authors must described the two species cultivation, application concentrations, and application methods.

We have moved Supplementary Materials and Methods to the main text and added more detailed descriptions in Materials and Methods. Also, to improve the logical flow and readability of our manuscript, we have added brief explanations about what/how we did at the first sentence of Results subsections (some of these explanations were only in Materials and Methods in the original manuscript). We have added the reference for how to cultivate G. nunn in L608 (Kobayashi et al., 2010; Tojo et al., 1993) (C. kiiensis was not cultivated but removed from the system as in Materials and Methods), and application concentrations. Application methods were described in Materials and Methods, the section Field manipulation experiments in 2019 in L596.

Comment #1-6 5. Validation: Application of one species clearly resulted to promote rice growth. They must include appropriate control treatment. If they pick same genus but different species that identified no specific effect on rice growth through eDNA analysis, no effect on growth can be provided. Generally application of large population of certain non-harmful organism confer plant growth promotion. It is not surprising result. Authors need to prove effectiveness of eDNA analysis. In addition, the field experiments required at least two years of consistent data for publication because environmental factors are so dynamic.

Thank you for pointing this out. We agree with your comment that species that were predicted to have no effect should not promote rice growth in a validation experiment. It was also one of our inititial experimental plans to include such species in our manipulation experiment in 2019, but we could not include them because of the limitation of time, labor, and money. More extensive validation of the statistical results of the 2017 data, including multi-year experiments, would further validate the effectiveness of our approach, which should be done as future studies. To clarify this point, we have added statements in the paragraph starting at L396.

Comment #1-7 In conclusion, I suggest that authors need more large data analysis and validate with more accurate and meaningful protocol.

As we explained in the revised manuscript and the Response to Comments #1-2 to #1-7, our study demonstrated a novel research framework to detect previously overlooked influential organisms under field conditions. We agree that larger data analysis would be ideal to further validate our approach, but whether and how to collect larger data is constrained by time, money, and labor. We believe that our study was designed carefully and could provide meaningful avenues for developing an ecological-network based, novel, and environment-friendly agriculture solutions.

**Reviewer #2 (Recommendations For The Authors):**
Comment #2-1 Lines 97-110: This is so cool. Modeling with empirical data is very powerful. But a rice field is an open system consisting of metacommunity dynamics. Maybe a tiny bit of biological and biogeochemical background here would be good.

Thank you for your comments. We have added a few examples of how and in which systems these methods were used to evaluate community dynamics and detect biological interactions in L109-L118.

Comment #2-2 Lines 111-126: I like the summary of the study here. I think the influential species concept can be a little more elevated. Paine's famous keystone species work has been cited but a couple more pieces of literature can help to enhance the ecological importance of this work.

We have explained the work by Paine (1966) a bit more and added one more paper that showed the effect of multiple predator species on the system dynamics at L88. We have also added a relevant sentence at L137 to emphasize the ecological/agricultural significance of our work.

Comment #2-3 Experimental design/Figure 1:Is there any rationale behind choosing red individuals to measure the growth?Is there any competition between the individuals in the pots?Figure 1e: It is nice to show the ASVs in time. I wonder how the plot would look like when normalized by biomass/DNA content/coverage/rarefaction because of the seasonality.

As for the first question, we chose the four individuals to minimize the edge effects (i.e., effects of microclimates and neighboring rice would be different between the four rice individuals and those planted in the edge regions). We have mentioned this in the legend of Figure 1.

As for the second question, there might be competition among the individuals in the pot. However, we did not measure the effect of competition (e.g., by comparing the growth with/without other rice individuals).

As for the third question, we published detailed dynamics of ecological community in the Supplementary Figures in Ushio (2022) Proceedings Bhttps://doi.org/10.6084/m9.figshare.c.5842766.v1. In addition, we haveuploaded a video showing the temporal dynamics of some top ( = most abundant) ASVs in https://doi.org/10.6084/m9.figshare.23514150.v2.

We have mentioned the supporting information in L153.

Comment #2-4 Line 146-147: Is this damage influence the inferences? Maybe it is better to justify.

While we occasionally observed physical damages, it is unlikely that they affected our causal inference because the changes in the rice heights due to the damages were smaller and less frequent than those due to growth. We have noted this at L151.

Comment #2-5 Line 161-162: Maybe refer readers to the methods section where you explain UIC analysis. It'd be easier to interpret the figures.

Mentioned.

Comment #2-6 Line 175-176: I believe very brief information in the intro about the organisms might help explain the hypothesis and interpret the results better.

We have included brief information of the two species at L197.

Comment #2-7 Figure 2: Species interaction strength: Are these proxies to the Jacobians? Is there a threshold for the influence we can consider strong/weak? For example, influential species compared to diagonal elements of the Jacobians (intraspecies interactions) could be shown as a mean vertical line in Figure 2b.

"Influences to rice growth" in Figure 2b is transfer entropy (TE) from a target ASV to rice growth. They are not proxies of the Jacobians, but they might positively correlate with the absolute value of the Jacobians. We have clarified this point in the legend (L953). More direct estimations of the Jacobian can be done using the MDR S-map method (Chang et al. 2021 DOI:10.1111/ele.13897), but we did not perform the MDR S-map in the present manuscript (see Ushio et al. 2023https://doi.org/10.7554/eLife.85795 for the application of the MDR S-map). As for TE, there is no clear threshold to distinguish strong/weak interactions.

Comment #2-8 Figure 2: Looking at panels c and d, it looks like there is a negative frequency selection between two influential species. Is it a reasonable observation?

This is an interesting point. In this manuscript, we have not carefully examined the interspecific relationship between these two particular species. However, the interspecific interactions were examined in detail and reported in Ushio (2022) Proceedings of the Royal Society B (DOI:10.1098/rspb.2021.2690). We re-checked the result in Ushio (2022); although there is a negative correlation between them, we did not find any (statistical) causal relationship between them.

Comment #2-9 Line 209: What is t-SNE analysis? Because of the manuscript's format, maybe methods should be shortly referred to in the relevant section or explained in brackets.

We have spelled out t-SNE.

Comment #2-10 Line 212-214: Maybe briefly explain what the hypotheses are for the alternative analysis, and what is the contribution of the results to the study.

We have added a brief explanation at L241: "Alternative statistical modeling that included the treatments (the control versus GN or CK treatments) and manipulation timing (i.e., before or after the manipulation), which simultaneously took the temporal changes of all the treatments into account, also showed qualitatively similar results (Supplementary file 4), further supporting the results."

Comment #2-11 Figure 3b/c: Maybe species names as panel titles could be helpful. d: Treatment names with initials in the legend could be also helpful to read the plots.

We have added species name as panel titles of Figure 3b,c. Treatment names were included in the legend of Figure 3.

Comment #2-12 Line 233: Maybe mention why the manuscript uses the word "clear".

We have mentioned this in L185.

Comment #2-13 Line 234-236: I think that these alternative tests should be explained somewhere.

We have revised the sentence so that it includes some explanations (L241). Also, we have referred to Materials and Methods.

Comment #2-14 Figure 4: The title says ecological community compositions, and panels show the growth rates and cumulative growth.

Thank you for pointing this out. This was a typo and we have corrected it.

Comment #2-15 Lines 246-269: Can these expression patterns be transient and relevant to the time point that the sample is taken?

Yes, these expression patterns were transient. We collected rice leaf samples for RNA-seq 1 day before the first manipulation and 1, 14, and 38 days after the third manipulation (see Supplementary file 3 for the sampling design). When we merged the pot locations, we observed no difference in the gene expression for samples 1 day before the first manipulation and 14 and 38 days after the third manipulation (except for two genes in samples 38 days after the manipulation), and thus, we consider the DEGs that appeared only in the short period after the manipulation. We have mentioned this in L278 and L383: "We found almost no DEGs for leaf samples taken one day before and 14 and 38 days after the third manipulation (the leaf sampling event 1, 3, and 4), suggesting that the influences of the treatments on the gene expression patterns were transient." (L278) and "These changes were observed relatively quickly and transient." (L383)

Comment #2-16 I wonder if a conceptual framework figure would help to generalize the workflow that can be used for other studies.

Thank you for your suggestion. Although we agree with your comment that such a figure would be helpful to generalize the workflow, we believe that our framework is clear and decided not to include it in the present manuscript. We might consider including such a figure (like Figure 1a in Ushio 2022) if we have an opportunity to write a review paper regarding this topic.

Comment #2-17 Lines 329-335: I feel this information is unclear in the early manuscript.Maybe it's necessary to clearly communicate in the beginning.

We have explained that we could not find any relevant information at least at the time we detected the ASVs in L189.

Comment #2-18 Lines 336-337: Can these species be identified in the previous data set from the ASV sequences?

Yes, these species were identified in the DNA data set obtained in 2017.

Comment #2-19 Lines 387-397: Are there any measurements such as total biomass, and statistical methods to help with the eDNA bias and data compositionality?

We have confirmed that our quantitative eDNA metabarcoding generates comparable results with the fluorescence-based method and quantitative PCR (e.g., see Supplementary Figures in Ushio 2022) (mentioned in L310 in the revised manuscript). However, at least in this study, we could not perform a direct comparison of the eDNA data with species abundance and/or biomass. This is partly because the number of our target species was too large (> 1,000 species). The accurate estimation of species abundance and/or biomass is one of our next goals.

Comment #2-20 Line 472: Maybe mention transfer entropy somewhere in the early manuscript.

We have mentioned this in L175.

Comment #2-21 Lines 494-503: Maybe a summary of this reasoning should be mentioned somewhere in the early manuscript too.

We have described a brief summary of the reasoning in L195.

Comment #2-22 Lines 29-33 If this sentence is simplified it might be easier to follow.

The sentence has been divided into two sentences in L28. Also, each sentence has been simplified.

Comment #2-23 Line 38 Maybe "macrobes" can be explicitly mentioned. Fungi, protozoa, etc.

Mentioned.

Comment #2-24 Line 139: I am not sure if the date should be in the title.

Similar monitoring was done in 2017 and 2019. Thus, we think the date is necessary in the section title.

Comment #2-25 Figure 1: There are 4 red individuals in the design but 5 measurements in the plots.

Heights and SPAD of the four individuals were measured for each plot and the averaged values were used as representative values for each plot. Therefore, 20 measurements ( = 4 rice individuals 5 plots) were done every day, but each plot has one rice height for each day. We have clarified this in the legend of Figure 1: "the average values of the four individuals were regarded as representative values for each plot."

Comment #2-26 Figure 1b: Maybe use the same axis length for the temperature as the other plots?

Corrected.

Comment #2-27 Lines 259-261: Are there the names of the genes in databases?

Yes, these are gene names used in the rice databases (e.g., The Rice Annotation Project Database; https://rapdb.dna.affrc.go.jp/inde x.html).

**Reviewer #3 (Recommendations For The Authors):**
Comment #3-1 Additionally, RGR is not statistically significant, but statistical significance is observed only in cumulative growth because data presentation does not reflect plant characteristics. RGR changes according to the developmental stage of the plant. Therefore, if RGR data are shown separately according to the rice growing season, the cumulative growth pattern and the pattern will appear similar.

RGRs were calculated daily (i.e., cm/day) and they changed depending on the developmental stage of the rice (Figure 1 and Figure 4–figure supplement 1). Therefore, we might find similar RGR patterns if we focus on a specific period of the growing season. However, unfortunately, we performed the intensive (i.e., daily) monitoring in 2019 only during the field manipulation period (middle June to middle July 2019), and we cannot investigate the changes in cumulative growth throughout the growing season (this depends on how many days we add up RGR to calculate the cumulative growth, though). We agree that, if we had investigated the detailed pattern of RGR throughout the growing season in 2019, we could have found similar pattens between RGR and cumulative growth rate at a certain period in the growing season. In Figure 4, the cumulative growths were calculated based on the RGRs before the third manipulation or during 10 days after the third manipulation. We clarified this in the legend of Figure 4.